# Forward and Backward Lifelong Learning with Time-dependent Tasks

## Abstract

For a sequence of classification tasks that arrive over time, lifelong learning methods can boost the effective sample size of each task by leveraging information from preceding and succeeding tasks (forward and backward learning). However, backward learning is often prone to a so-called catastrophic forgetting in which a task's performance gets worse while trying to repeatedly incorporate information from succeeding tasks. In addition, current lifelong learning techniques are designed for i.i.d. tasks and cannot capture the usual higher similarities between consecutive tasks. This paper presents lifelong learning methods based on minimax risk classifiers (LMRCs) that effectively exploit forward and backward learning and account for time-dependent tasks. In addition, we analytically characterize the increase in effective sample size provided by forward and backward learning in terms of the tasks' expected quadratic change. The experimental evaluation shows that LMRCs can result in a significant performance improvement, especially for reduced sample sizes.

## 1 Introduction

In practical scenarios, classification problems (tasks) often have limited sample sizes and arrive sequentially over time. Lifelong learning (also known as continual learning) can boost the effective sample size (ESS) of each task by leveraging information from preceding and succeeding tasks (forward and backward learning) (Ruvolo & Eaton, 2013; Lopez-Paz & Ranzato, 2017; Chen & Liu, 2018). The general goal of such approaches is to replicate the humans' ability to continually improve the performance of each task exploiting information acquired from other tasks.

The development of lifelong learning techniques is hindered by the continuous arrival of samples from tasks characterized by different underlying distributions. In particular, backward learning (also known as reverse transfer) is often prone to a so-called catastrophic forgetting in which a task's performance gets worse while trying to repeatedly incorporate information from the succeeding tasks (Kirkpatrick et al., 2017; Hurtado et al., 2021; Henning et al., 2021). More generally, lifelong learning methods face a so-called stability-plasticity dilemma: the excessive usage of information from different tasks can result in a performance decrease while a moderate usage does not fully exploit the potential of lifelong learning (Rolnick et al., 2019; Ke et al., 2021).

Most of lifelong learning techniques are designed for tasks sampled i.i.d. from a task environment (Baxter, 2000; Maurer et al., 2016; Denevi et al., 2019), and current methods cannot capture the usual higher similarities between consecutive tasks. For a sequence of tasks that arrive over time, it is common that the tasks are time-dependent and consecutive tasks are significantly more similar. For instance, if each task corresponds to the classification of portraits from a specific time period (Ginosar et al., 2015), the similarity between tasks is markedly higher for consecutive tasks (see Figure 1). In the current literature of lifelong learning, only Pentina & Lampert (2015) considers scenarios with time-dependent tasks and analyzes the feasibility of transferring information from the preceding tasks. On the other hand, methods designed for concept drift adaptation (Zhao et al., 2020; Tahmasbi et al., 2021; Álvarez et al., 2022) account for time-dependent underlying distributions but only aim to learn the last task in the sequence.

This paper presents lifelong learning methods based on minimax risk classifiers (LMRCs). The proposed techniques effectively exploit forward and backward learning and account for time-dependent tasks. Specifically, the main contributions presented in the paper are as follows.

Backward

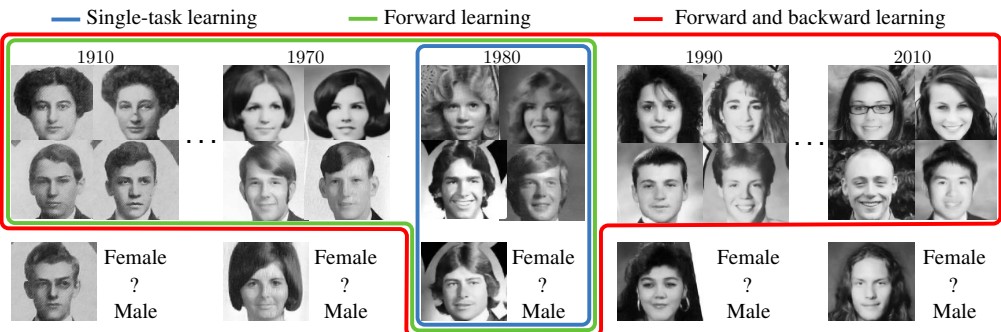

Figure 1: For tasks that arrive over time, consecutive tasks are often more similar. Forward and backward learning can exploit such similarities and extract information from preceding and succeding tasks.

- The presented LMRCs minimize the worst-case error probabilities over uncertainty sets obtained using information from all the tasks.
- We propose learning techniques that can effectively incorporate information from the ever-increasing sequence of tasks and provide performance guarantees for forward and backward learning.
- We analytically characterize the increase in ESS provided by forward and backward learning in terms of the expected quadratic change between consecutive tasks.
- We numerically quantify the performance improvement provided by the presented learning techniques in comparison with existing methods using multiple datasets, different sample sizes, and number of tasks.

**Notations** Calligraphic letters represent sets; $\|\cdot\|_1$ and $\|\cdot\|_\infty$ denote the 1-norm and the infinity norm of its argument, respectively; $\preceq$ and $\succeq$ denote vector inequalities; $\mathbb{I}\{\cdot\}$ denotes the indicator function; and $\mathbb{E}_p\{\cdot\}$ and $\mathbb{V}\mathrm{ar}_p\{\cdot\}$ denote the expectation and the variance of its argument with respect to distribution p. For a vector $\mathbf{v}$, $v^{(i)}$ and $\mathbf{v}^T$ denote the i-th component and the transpose of $\mathbf{v}$. Non-linear operators acting on vectors denote component-wise operations. For instance, $|\mathbf{v}|$ and $\mathbf{v}^2$ denote the vector formed by the absolute value and the square of each component, respectively.

## 2 PRELIMINARIES

In the following, we denote by $\mathcal{X}$ the set of instances or attributes, $\mathcal{Y}$ the set of labels or classes, $\Delta(\mathcal{X} \times \mathcal{Y})$ the set of probability distributions over $\mathcal{X} \times \mathcal{Y}$, and $\mathrm{T}(\mathcal{X}, \mathcal{Y})$ the set of classification rules. A classification task is characterized by an underlying distribution $p^* \in \Delta(\mathcal{X} \times \mathcal{Y})$ and supervised classification methods use a sample set $D = \{(x_i, y_i)\}_{i=1}^n$ formed by $n$ i.i.d samples from distribution $p^*$ to find a classification rule $\mathrm{h} \in \mathrm{T}(\mathcal{X}, \mathcal{Y})$ with small expected loss $\ell(\mathrm{h}, p^*)$.

In lifelong learning, sample sets $D_1, D_2, \ldots$ arrive over time steps $1, 2, \ldots$ corresponding with different classification tasks characterized by underlying distributions $p_1, p_2, \ldots$. At each time step $k$, lifelong learning methods aim to obtain classification rules $h_1, h_2, \ldots, h_k$ with small expected losses $\ell(h_1, p_1), \ell(h_2, p_2), \ldots, \ell(h_k, p_k)$ for the current sequence of $k$ tasks. For instance, overall performance is usually assessed by the averaged error $\frac{1}{k}\sum_{i=1}^k \ell(h_i, p_i)$. As depicted in Fig. 1, for each $j$-th task with $j \in \{1, 2, \ldots, k\}$, lifelong learning methods obtain the classification rule $h_j$ leveraging information obtained from sample sets $D_1, D_2, \ldots, D_j$ (forward learning) and from sample sets $D_{j+1}, D_{j+2}, \ldots, D_k$ (backward learning). Most existing lifelong learning techniques are designed for tasks characterized by distributions $p_1, p_2, \ldots$ such that the tasks' distributions $p_i$ are independent and identically distributed (i.i.d.) random probability measures for $i = 1, 2, \ldots$. In the following, we propose lifelong learning techniques designed for time-dependent tasks that are characterized by distributions $p_1, p_2, \ldots$ such that the changes between consecutive distributions $p_{i+1} - p_i$ are independent and zero-mean random signed measures for $i = 1, 2, \ldots$. Such assumption can account for usual higher similarities between consecutive tasks; for instance, it implies that $p_{i+t} - p_i$ is a zero-mean random variable with $\mathbb{V}\mathrm{ar}\{p_{i+t} - p_i\} = \sum_{j=1}^t \mathbb{V}\mathrm{ar}\{p_{i+j} - p_{i+j-1}\}$, while

the i.i.d. case would imply that $p_{i+t} - p_i$ is a zero-mean random variable with $\mathbb{V}\text{ar}\{p_{i+t} - p_i\} = \mathbb{V}\text{ar}\{p_{i+1} - p_i\} = 2\mathbb{V}\text{ar}\{p_1\}$ for any $t$ and $i$.

As described above, lifelong learning methods consider tasks characterized by different distributions. The methods presented below are based on the framework of minimax risk classifiers (MRCs) (Mazuelas et al., 2020; 2022a;b) since MRCs can utilize general expectation estimates obtained from samples with different distributions. MRCs learn classification rules by minimizing the worst-case expected loss against an uncertainty set that can include the underlying distribution with high probability. Such uncertainty sets are given by constraints on the expectation of a feature mapping $\Phi : \mathcal{X} \times \mathcal{Y} \to \mathbb{R}^m$ as

$$\mathcal{U} = \{p \in \Delta(\mathcal{X} \times \mathcal{Y}) : |\mathbb{E}_p\{\Phi(x,y)\} - \boldsymbol{\tau}| \preceq \boldsymbol{\lambda}\} \tag{1}$$

where $\boldsymbol{\tau}$ denotes a mean vector of expectation estimates and $\boldsymbol{\lambda}$ denotes a confidence vector. Feature mappings are vector-valued functions over $\mathcal{X} \times \mathcal{Y}$, e.g., one-hot encodings of values from the last layers in a neural network (Bengio et al., 2013; Kemker & Kanan, 2018; Mohri et al., 2018; Hurtado et al., 2021).

Given the uncertainty set $\mathcal{U}$, MRC rules are solutions of the optimization problem

$$R(\mathcal{U}) = \min_{h \in T(\mathcal{X}, \mathcal{Y})} \max_{p \in \mathcal{U}} \ell(h, p) \tag{2}$$

where $R(\mathcal{U})$ denotes the minimax risk and $\ell(h, p)$ denotes the expected loss of classification rule h for distribution p. In the following, we utilize the 0-1-loss so that $\ell(h, p) = \mathbb{E}_p\{\mathbb{I}\{h(x) \neq y\}\}$ and the expected loss with respect to the underlying distribution becomes the error probability of the classification rule. Deterministic MRCs assign each instance $x \in \mathcal{X}$ with the label $h(x) \in \arg\max_{y \in \mathcal{Y}} \Phi(x, y)^{\mathrm{T}} \boldsymbol{\mu}^*$ where the parameter $\boldsymbol{\mu}^*$ is the solution of the convex optimization problem

$$\min_{\boldsymbol{\mu}} 1 - \boldsymbol{\tau}^{\mathrm{T}} \boldsymbol{\mu} + \max_{x \in \mathcal{X}, \mathcal{C} \subseteq \mathcal{Y}} \frac{\sum_{y \in \mathcal{C}} \Phi(x, y)^{\mathrm{T}} \boldsymbol{\mu} - 1}{|\mathcal{C}|} + \boldsymbol{\lambda}^{\mathrm{T}} |\boldsymbol{\mu}| \tag{3}$$

given by the Lagrange dual of (2).

MRCs provide bounds for the error probability with respect to the minimax risk $R(\mathcal{U})$ and the smallest minimax risk as described in Mazuelas et al. (2020; 2022a;b). The smallest minimax risk, denoted by $R^{\infty}$, is the minimax risk corresponding to the ideal case of knowing mean vectors exactly, that is, $R^{\infty}$ is the minimax risk corresponding with the uncertainty set $\mathcal{U}^{\infty} = \{p \in \Delta(\mathcal{X} \times \mathcal{Y}) : \mathbb{E}_p\{\Phi(x,y)\} = \mathbb{E}_{p^*}\{\Phi(x,y)\}\}$. Such minimax risk coincides with the Bayes risk if the underlying distribution is the worst-case distribution in uncertainty set given by the true expectation of feature mapping (Mazuelas et al., 2022a). If the mean vector of expectation estimates is obtained using a sample set $D$ formed by instance-label pairs from the same underlying distribution, then the performance bounds of MRCs are of the usual order $\mathcal{O}(1/\sqrt{n})$ where $n$ is the sample size of $D$.

In lifelong learning, the baseline approach of single-task learning obtains a classification rule $h_j$ for each $j$-th task leveraging information only from the sample set $D_j = \{(x_{j,i}, y_{j,i})\}_{i=1}^{n_j}$ of size $n_j$. In that case, LMRCs coincide with MRCs for standard supervised classification that obtain the mean and confidence vectors as

$$\boldsymbol{\tau}_j = \frac{1}{n_j} \sum_{i=1}^{n_j} \Phi(x_{j,i}, y_{j,i}), \quad \boldsymbol{\lambda}_j = \sqrt{\boldsymbol{s}_j}, \quad \boldsymbol{s}_j = \boldsymbol{\sigma}_j^2 / n_j \tag{4}$$

with $\boldsymbol{\sigma}_j^2$ an estimate of $\mathbb{V}\text{ar}_{p_j}\{\Phi(x,y)\}$, e.g., the sample variance of the $n_j$ samples. The vector $\boldsymbol{s}_j$ describes the mean squared errors (MSEs) of the mean vector components and directly gives the confidence vector $\boldsymbol{\lambda}_j$ as shown in (4).

In the following sections, we describe techniques that obtain the mean and MSE vectors using forward and backward learning. Once such vectors are obtained, LMRC methods take the confidence vector $\boldsymbol{\lambda}_j$ as in (4) and obtain the classifier parameter $\boldsymbol{\mu}_j$ for each $j$-th task solving the convex optimization problem in (3) that can be efficiently addressed using conventional methods (Nesterov & Shikhman, 2015; Tao et al., 2019).

## 3 Forward Learning with Performance Guarantees

This section presents the recursions that allow to obtain mean and MSE vectors for each task retaining information from preceding tasks. In addition, it characterizes the increase in ESS provided by forward learning in terms of the tasks' expected quadratic change and the number of tasks.

### 3.1 Forward Learning

The proposed techniques for forward learning account for time-dependent tasks and obtain classification rules for each task leveraging information from preceding tasks. Let $\vec{\tau_j}$ and $\vec{s_j}$ denote the mean and MSE vectors for forward learning corresponding to the $j$-th task for $j \in \{1, 2, \ldots, k\}$. The following recursions allow to obtain $\vec{\tau_j}$ and $\vec{s_j}$ for each $j$-th task using those vectors for the preceding task $\vec{\tau}_{j-1}, \vec{s}_{j-1}$ as

$$\vec{\tau_j} = \tau_j + \frac{s_j}{\vec{s}_{j-1} + s_j + d_j^2} \left( \vec{\tau}_{j-1} - \tau_j \right) \tag{5}$$

$$\vec{s_j} = \left( \frac{1}{s_j} + \frac{1}{\vec{s}_{j-1} + d_j^2} \right)^{-1} \tag{6}$$

with $\tau_j$ and $s_j$ given by (4) and $\vec{\tau_1} = \tau_1$, $\vec{s_1} = s_1$.

The vector $d_j^2$ assesses the expected quadratic change between consecutive tasks. In the following, the change between consecutive tasks is described by $w_j = \tau_j^\infty - \tau_{j-1}^\infty$ for any $j \in \{2, 3, \ldots, k\}$ where $\tau_j^\infty = \mathbb{E}_{p_j}\{\Phi(x, y)\}$ is the expectation of the feature mapping with respect to the underlying distribution. If $p_j - p_{j-1}$ are independent and zero-mean for $j = 2, 3, \ldots$, then vectors $w_j$ are also independent and zero-mean for any feature mapping.

Taking $d_i^2 = \mathbb{E}\{w_i^2\} = \mathbb{E}\{(\tau_i^\infty - \tau_{i-1}^\infty)^2\}$ and $\sigma_i^2 = \mathbb{V}\mathrm{ar}_{p_i}\{\Phi(x, y)\}$ for any $i$, the recursion in (5) provides the unbiased linear estimator of the mean vector $\tau_j^\infty$ based on $D_1, D_2, \ldots, D_j$ that has the minimum MSE, while the recursion in (6) provides its MSE (see Appendix A for a detailed derivation). Vectors $\sigma_i^2$ and $d_i^2$ can be estimated online using the sample sets. In particular, $\sigma_i^2$ can be estimated as the sample variance, while $d_i^2$ can be estimated using sample averages as

$$d_i^2 = \frac{1}{W} \sum_{l=1}^{W} (\tau_{i_l} - \tau_{i_{l-1}})^2 \tag{7}$$

where $i_0, i_1, \ldots, i_W$ are the $W + 1$ closest indexes to $i$ in $\{1, 2, \ldots, k\}$.

Recursions (5)-(6) obtain mean and MSE vectors for the $j$-th task by acquiring information from the $j$-th sample set $D_j$ and retaining information from preceding tasks. Specifically, recursion (5) obtains the mean vector $\vec{\tau_j}$ by adding a correction to the sample average $\tau_j$. This correction is proportional to the difference between $\tau_j$ and $\vec{\tau}_{j-1}$ with a proportionality constant that depends on the MSE vectors $s_j, \vec{s}_{j-1}$ and the expected quadratic change $d_j^2$. In particular, if $s_j \ll \vec{s}_{j-1} + d_j^2$, the mean vector is given by the sample average as in single-task learning, and if $s_j \gg \vec{s}_{j-1} + d_j^2$, the mean vector is given by that of the preceding task. Note that for forward learning, at each step $k$, only the vectors for the last task $\vec{\tau_k}$ and $\vec{s_k}$ need to be obtained from those of the $(k-1)$-th task. The vectors for the remaining $j$-th tasks with $j \in \{1, 2, \ldots, k-1\}$ stay the same as at step $k-1$ (see also Fig. 2 and Alg. 1 below).

### 3.2 Performance Guarantees and Effective Sample Sizes

The following result provides bounds for the minimax risk for each task with respect to the smallest minimax risk. For each $j$-th task, we denote by $R_j^\infty$ the smallest minimax risk and by $\mu_j^\infty$ the classifier parameter that determines the optimal minimax rule, as described in Section 2. In addition, we denote by $R(\vec{\mathcal{U}_j})$ the minimax risk over uncertainty set $\vec{\mathcal{U}_j}$ determined as in (1) using the mean and confidence vectors $\vec{\tau_j}$ and $\vec{\lambda_j} = \sqrt{\vec{s_j}}$ provided by (5) and (6).

**Theorem 1.** Let $M$ and $\kappa$ be such that $M \geq \|\Phi(x,y)\|_\infty \; \forall (x,y) \in \mathcal{X} \times \mathcal{Y}$ and

$$\kappa \geq \frac{\sigma\left(\Phi_j^{(i)}\right)}{\sigma_j^{(i)}}, \kappa \geq \frac{\sigma\left(w_j^{(i)}\right)}{d_j^{(i)}} \text{ for } j = 1, 2, \ldots, k \text{ and } i = 1, 2, \ldots, m$$

where $\Phi_j^{(i)}$ denotes the r.v. given by the $i$-th component of the feature mapping of samples from the $j$-th task, $\sigma(z)$ denotes the sub-Gaussian parameter of a r.v. $z$, i.e., $\mathbb{E}\{e^{t(z - \mathbb{E}\{z\})}\} \leq e^{\sigma(z)^2 t^2 / 2} \; \forall t$. For any $j \in \{1, 2, \ldots, k\}$, we have that

$$R(\mathcal{U}_j^{\rightarrow}) \leq R_j^\infty + \frac{M(\kappa + 1)\sqrt{2\log(2m/\delta)}}{\sqrt{n_j^{\rightarrow}}} \left\|\boldsymbol{\mu}_j^\infty\right\|_1 \text{ with prob. at least } 1 - \delta \quad (8)$$

with $n_1^{\rightarrow} = n_1$ and $n_j^{\rightarrow} \geq n_j + n_{j-1}^{\rightarrow} \frac{\|\boldsymbol{\sigma}_j^2\|_\infty}{\|\boldsymbol{\sigma}_j^2\|_\infty + n_{j-1}^{\rightarrow}\|\boldsymbol{d}_j^2\|_\infty}$ for $j \geq 2$.

*Proof.* See Appendix B. $\qquad\square$

The excess risk in inequality (8) decreases as $\mathcal{O}(1/\sqrt{n_j^{\rightarrow}})$ using the forward learning methods proposed, while such difference would decrease as $\mathcal{O}(1/\sqrt{n_j})$ using only the information of the $j$-th task. Therefore, $n_j^{\rightarrow}$ in (8) is the ESS of the proposed LMRC method with forward learning. The ESS of each task is obtained by adding a fraction of the ESS for the preceding task to the sample size. In particular, if $\boldsymbol{d}_j^2$ is large, the ESS is given by the sample size, while if $\boldsymbol{d}_j^2$ is small, the ESS is given by the sum of the sample size and the ESS of the preceding task.

Other existing methods provide comparable performance bounds (Mohri & Medina, 2012; Pentina & Lampert, 2015). Such bounds decrease with the number of tasks and increase with the change between consecutive distributions. Specifically, bounds in Proposition 1 of Mohri & Medina (2012) and in Theorem 7 of Pentina & Lampert (2015) are proportional to the discrepancy and to the Kullback-Leibler divergence between consecutive distributions. The bound in Theorem 1 above decreases with the number of tasks and increases with the expected quadratic change $\boldsymbol{d}_j^2$ between consecutive distributions. Note that the coefficient $\kappa$ in (8) can be taken to be small as long as the values used for $\sigma_j$ and $d_j$ are not much lower than the sub-Gaussian parameters of $\Phi_j$ and $w_j$, respectively. In particular, $\kappa$ is smaller than the maximum of $M/\min_{j,i}\{\sigma_j^{(i)}\}$ and $2M/\min_{j,i}\{d_j^{(i)}\}$ due to the bound for the sub-Gaussian parameter of bounded random variables (see e.g., Section 2.1.2 in Wainwright (2019)).

Theorem 1 shows the increase in ESS in terms of the ESS of the preceding task. The following result allows to directly quantify the ESS in terms of the sample size and the expected quadratic change.

**Theorem 2.** Let $d$, $\boldsymbol{\sigma}_j$ and $n$ be such that $d^2 \geq \|\boldsymbol{d}_j^2\|_\infty$, $\|\boldsymbol{\sigma}_j^2\|_\infty \leq 1$, and $n \leq n_j$ for $j = 1, 2, \ldots, k$. For any $j \in \{1, 2, \ldots, k\}$, we have that the ESS in (8) can be taken so that it satisfies

$$n_j^{\rightarrow} \geq n\left(1 + \frac{(1+\alpha)^{2j-1} - 1 - \alpha}{\alpha(1+\alpha)^{2j-1} + \alpha}\right) \text{ with } \alpha = \frac{2}{\sqrt{1 + \frac{4}{nd^2}} - 1}. \quad (9)$$

In particular, for $j \geq 2$, we have that

$$n_j^{\rightarrow} \geq n\left(1 + \frac{j-1}{3}\right) \qquad \text{if} \qquad nd^2 < \frac{1}{j^2}$$

$$n_j^{\rightarrow} \geq n\left(1 + \frac{1}{5\sqrt{nd^2}}\right) \qquad \text{if} \qquad \frac{1}{j^2} \leq nd^2 < 1$$

$$n_j^{\rightarrow} \geq n\left(1 + \frac{1}{3nd^2}\right) \qquad \text{if} \qquad nd^2 \geq 1.$$

*Proof.* See Appendix C. $\qquad\square$

The above theorem characterizes the increase in ESS provided by forward learning in terms of the tasks' expected quadratic change. Such increase grows monotonically with the number of preceding tasks $j$ as shown in (9) and becomes proportional to $j$ when the expected quadratic change is smaller than $1/(j^2 n)$. Figure 3 below further illustrates the increase in ESS with respect to the sample size $(n_j^{\rightarrow}/n)$ due forward learning in comparison with forward and backward learning.

## 4 FORWARD AND BACKWARD LEARNING WITH PERFORMANCE GUARANTEES

This section presents the recursions that allow to obtain mean and MSE vectors for each task retaining information from preceding tasks and acquiring information from succeeding tasks. In addition, it characterizes the increase in ESS provided by forward and backward learning in terms of the tasks' expected quadratic change and the number of tasks.

Backward learning is more challenging than forward learning since, for each task, the sequence of succeeding tasks is ever-increasing due to the continuous arrival of tasks, while the sequence of preceding tasks is always the same. The repeated usage of information from the succeeding tasks can result in a so-called catastrophic forgetting in which the tasks' performance gets worse over time. The techniques proposed below for backward learning effectively increase the ESS over time by carefully accounting for the new information at each step.

### 4.1 FORWARD AND BACKWARD LEARNING

The proposed techniques for forward and backward learning account for time-dependent tasks and obtain classification rules for each task leveraging information from preceding and succeeding tasks. From preceding tasks, we obtain the forward mean and MSE vectors $\boldsymbol{\tau}_j^{\rightarrow}, \boldsymbol{s}_j^{\rightarrow}$ using recursions (5)-(6), while from succeeding tasks, we obtain the backward mean and MSE vectors $\boldsymbol{\tau}_j^{\leftarrow k}, \boldsymbol{s}_j^{\leftarrow k}$ using recursions (5)-(6) in retrodiction. Specifically, vectors $\boldsymbol{\tau}_j^{\leftarrow k}$ and $\boldsymbol{s}_j^{\leftarrow k}$ are obtained using the same recursion as for $\boldsymbol{\tau}_j^{\rightarrow}$ and $\boldsymbol{s}_j^{\rightarrow}$ in (5)-(6) with $\boldsymbol{s}_{j+1}^{\leftarrow k}, \boldsymbol{d}_{j+1}^2$, and $\boldsymbol{\tau}_{j+1}^{\leftarrow k}$ instead of $\boldsymbol{s}_{j-1}^{\rightarrow}, \boldsymbol{d}_j^2$, and $\boldsymbol{\tau}_{j-1}^{\rightarrow}$.

Let $\boldsymbol{\tau}_j^{\rightleftharpoons k}$ and $\boldsymbol{s}_j^{\rightleftharpoons k}$ denote the mean and MSE vectors for forward and backward learning corresponding to the $j$-th task for $j \in \{1, 2, \ldots, k\}$. The following recursions allow to obtain, at each step $k$, the mean and MSE vectors $\boldsymbol{\tau}_j^{\rightleftharpoons k}$ and $\boldsymbol{s}_j^{\rightleftharpoons k}$ for each $j$-th task using those vectors for forward learning $\boldsymbol{\tau}_j^{\rightarrow}, \boldsymbol{s}_j^{\rightarrow}$ and backward learning $\boldsymbol{\tau}_{j+1}^{\leftarrow k}, \boldsymbol{s}_{j+1}^{\leftarrow k}$ as $\boldsymbol{\tau}_k^{\rightleftharpoons k} = \boldsymbol{\tau}_k^{\rightarrow}, \boldsymbol{s}_k^{\rightleftharpoons k} = \boldsymbol{s}_k^{\rightarrow}$ and

$$\boldsymbol{\tau}_j^{\rightleftharpoons k} = \boldsymbol{\tau}_j^{\rightarrow} + \frac{\boldsymbol{s}_j^{\rightarrow}}{\boldsymbol{s}_j^{\rightarrow} + \boldsymbol{s}_{j+1}^{\leftarrow k} + \boldsymbol{d}_{j+1}^2} \left( \boldsymbol{\tau}_{j+1}^{\leftarrow k} - \boldsymbol{\tau}_j^{\rightarrow} \right) \tag{10}$$

$$\boldsymbol{s}_j^{\rightleftharpoons k} = \left( \frac{1}{\boldsymbol{s}_j^{\rightarrow}} + \frac{1}{\boldsymbol{s}_{j+1}^{\leftarrow k} + \boldsymbol{d}_{j+1}^2} \right)^{-1} \tag{11}$$

with $\boldsymbol{\tau}_k^{\leftarrow k} = \boldsymbol{\tau}_k$ and $\boldsymbol{s}_k^{\leftarrow k} = \boldsymbol{s}_k$. Analogously to the case of forward learning in Section 3.1, taking $\boldsymbol{d}_i^2 = \mathbb{E}\{\boldsymbol{w}_i^2\}$ and $\boldsymbol{\sigma}_i^2 = \mathbb{V}\mathrm{ar}_{\mathrm{p}_i}\{\Phi(x, y)\}$ for any $i$, the recursion in (10) provides the unbiased linear estimator of the mean vector $\boldsymbol{\tau}_j^{\infty}$ based on $D_1, D_2, \ldots, D_j$ and $D_{j+1}, D_{j+2}, \ldots, D_k$ that has the minimum MSE, while the recursion in (11) provides its MSE (see Appendix A for a detailed derivation).

Recursions (10)-(11) obtain at step $k$ the mean and MSE vectors for the $j$-th task by retaining information from preceding tasks and acquiring information from succeeding tasks. Specifically, recursion (10) obtains the mean vector $\boldsymbol{\tau}_j^{\rightleftharpoons k}$ by adding a correction to the mean vector of the corresponding task $\boldsymbol{\tau}_j^{\rightarrow}$ obtained for forward learning. This correction is proportional to the difference between $\boldsymbol{\tau}_j^{\rightarrow}$ and $\boldsymbol{\tau}_{j+1}^{\leftarrow k}$ with a proportionality constant that depends on the MSE vectors $\boldsymbol{s}_j^{\rightarrow}, \boldsymbol{s}_{j+1}^{\leftarrow k}$ and the expected quadratic change $\boldsymbol{d}_{j+1}^2$. In particular, if $\boldsymbol{s}_j^{\rightarrow} \ll \boldsymbol{s}_{j+1}^{\leftarrow k} + \boldsymbol{d}_{j+1}^2$, the mean vector is given by that of the corresponding task for forward learning, and if $\boldsymbol{s}_j^{\rightarrow} \gg \boldsymbol{s}_{j+1}^{\leftarrow k} + \boldsymbol{d}_{j+1}^2$, the mean vector is given by that of the succeeding task for backward learning.

### 4.2 IMPLEMENTATION

This section describes the implementation of the proposed LMRCs with forward and backward learning and its computational and memory complexities.

Figure 2 depicts the flow diagram for the proposed LMRC methodology. The proposed techniques carefully avoid the repeated usage of the same information from the sequence of succeeding tasks.

At each step $k$, new backward mean vectors $\boldsymbol{\tau}_{j+1}^{\leftharpoonup k}$ for $j = k-1, k-2, .., k-b$ are obtained for $b$ backward steps, then the forward and backward mean vectors $\boldsymbol{\tau}_j^{\rightleftharpoons k}$ given by (10) are obtained from the forward mean vectors $\boldsymbol{\tau}_j^{\rightharpoonup}$ and the backward mean vectors $\boldsymbol{\tau}_{j+1}^{\leftharpoonup k}$. In particular, $\boldsymbol{\tau}_j^{\rightharpoonup}$ provides the information from the preceding tasks $1, 2, \ldots, j$, while $\boldsymbol{\tau}_{j+1}^{\leftharpoonup k}$ provides the information from the succeeding tasks $j+1, j+2, \ldots, k$.

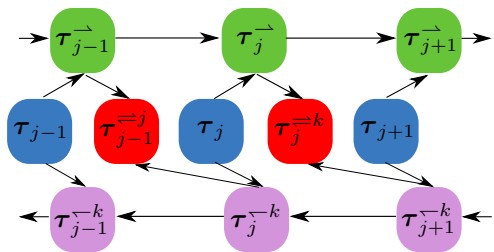

Figure 2: Diagram for LMRC methodology.

Algorithm 1 details the implementation of the proposed LMRCs at each step. For $k$ steps, LMRCs have computational complexity $\mathcal{O}((b+1)Kmk)$ and memory complexity $\mathcal{O}((b+k)m)$ where $K$ is the number of iterations used for the convex optimization problem (3), $m$ is the length of the feature vector, and $b$ is the number of backward steps. In particular, if $b=0$, LMRC carries out only forward learning. The complexity of forward and backward learning increases proportionally to the number of backward steps that can be taken to be rather small, as shown in the following. Even more efficient implementations can be obtained using Rauch-Tung-Striebel recursions (see e.g., Section 7.2 in Anderson & Moore (1979)) that can obtain $\boldsymbol{\tau}_j^{\rightleftharpoons k}$ from $\boldsymbol{\tau}_{j+1}^{\rightleftharpoons k}$ as shown in Appendix D.

---

**Algorithm 1** LMRC at step $k$

---

**Input:** $D_k$ from new task and $\boldsymbol{\tau}_j, \boldsymbol{s}_j, \boldsymbol{\tau}_j^{\rightharpoonup}, \boldsymbol{s}_j^{\rightharpoonup}$ for $k-b \leq j < k$ from previous $b-1$ steps
**Output:** $\boldsymbol{\mu}_j$ for $k-b \leq j \leq k, \boldsymbol{\tau}_k, \boldsymbol{s}_k, \boldsymbol{\tau}_k^{\rightharpoonup}, \boldsymbol{s}_k^{\rightharpoonup}$
Obtain sample average and MSE vectors $\boldsymbol{\tau}_k^{\leftharpoonup k} = \boldsymbol{\tau}_k, \boldsymbol{s}_k^{\leftharpoonup k} = \boldsymbol{s}_k$ using the sample set $D_k$ ▷ Single-task
Estimate the tasks' expected quadratic change $\boldsymbol{d}_k^2$ using (7)
Obtain the forward mean and MSE vectors $\boldsymbol{\tau}_k^{\rightharpoonup}, \boldsymbol{s}_k^{\rightharpoonup}$ using (5)-(6) ▷ Forward
Take $\boldsymbol{\lambda}_k^{\rightharpoonup} = \sqrt{\boldsymbol{s}_k^{\rightharpoonup}}$ and obtain classifier parameter $\boldsymbol{\mu}_k$ solving the optimization problem (3)
**for** $j = k-1, k-2, \ldots, k-b$ **do**
    Estimate the tasks' expected quadratic change $\boldsymbol{d}_j^2$ using (7)
    Obtain backward mean and MSE vectors $\boldsymbol{\tau}_{j+1}^{\leftharpoonup k}, \boldsymbol{s}_{j+1}^{\leftharpoonup k}$ using (5)-(6) in retrodiction ▷ Backward
    Obtain mean and MSE vectors $\boldsymbol{\tau}_j^{\rightleftharpoons k}, \boldsymbol{s}_j^{\rightleftharpoons k}$ using (10)-(11) ▷ Forward and backward
    Take $\boldsymbol{\lambda}_j^{\rightleftharpoons k} = \sqrt{\boldsymbol{s}_j^{\rightleftharpoons k}}$ and obtain classifier parameters $\boldsymbol{\mu}_j$ solving the optimization problem (3)

---

### 4.3 PERFORMANCE GUARANTEES AND EFFECTIVE SAMPLE SIZES

The following result provides bounds for the minimax risk for each task with respect to the smallest minimax risk. For each $j$-th task and step $k$, we denote by $R(\mathcal{U}_j^{\rightleftharpoons k})$ the minimax risk over uncertainty set $\mathcal{U}_j^{\rightleftharpoons k}$ determined as in (1) using the mean and confidence vector $\boldsymbol{\tau}_j^{\rightleftharpoons k}$ and $\boldsymbol{\lambda}_j^{\rightleftharpoons k} = \sqrt{\boldsymbol{s}_j^{\rightleftharpoons k}}$ provided by (10) and (11).

**Theorem 3.** Let $M, \kappa$, and $n_j^{\rightharpoonup}$ be as in Theorem 1. For any $j \in \{1, 2, \ldots, k\}$, we have that

$$R(\mathcal{U}_j^{\rightleftharpoons k}) \leq R_j^\infty + \frac{M(\kappa+1)\sqrt{2\log(2m/\delta)}}{\sqrt{n_j^{\rightleftharpoons k}}} \left\| \boldsymbol{\mu}_j^\infty \right\|_1 \text{ with prob. at least } 1-\delta \qquad (12)$$

with $n_k^{\rightleftharpoons k} = n_k^{\rightharpoonup}$ and $n_j^{\rightleftharpoons k} \geq n_j^{\rightharpoonup} + n_{j+1}^{\leftharpoonup k} \frac{\|\boldsymbol{\sigma}_j^2\|_\infty}{\|\boldsymbol{\sigma}_j^2\|_\infty + n_{j+1}^{\leftharpoonup k}\|\boldsymbol{d}_{j+1}^2\|_\infty}$ for $j \leq k-1$, where the backward ESSs satisfy $n_k^{\leftharpoonup k} = n_k$ and $n_j^{\leftharpoonup k} \geq n_j + n_{j+1}^{\leftharpoonup k} \frac{\|\boldsymbol{\sigma}_j^2\|_\infty}{\|\boldsymbol{\sigma}_j^2\|_\infty + n_{j+1}^{\leftharpoonup k}\|\boldsymbol{d}_{j+1}^2\|_\infty}$.

*Proof.* See Appendix E. $\qquad\qquad\square$

To the best of our knowledge, Theorem 3 provides the first performance guarantees for lifelong learning that show positive backward transfer. In particular, the bounds for forward and backward learning provided by inequality (12) are significantly lower than those for forward learning in Theorem 1. The ESS of each task is obtained by adding a fraction of the ESS for the succeeding task to the ESS of the corresponding task using forward learning. In particular, if $\boldsymbol{d}_j^2$ is large, the ESS is given by that with forward learning, while if $\boldsymbol{d}_j^2$ is small, the ESS is given by the sum of the ESS using forward learning and the ESS of the succeeding task.

Theorem 3 shows the increase in ESS in terms of the ESS with forward learning and the ESS of the succeeding task. The following result allows to directly quantify the ESS in terms of the sample size and the expected quadratic change.

**Theorem 4.** Let $d$, $\boldsymbol{\sigma}_j$ and $n$ be such that $d^2 \geq \|\boldsymbol{d}_j^2\|_\infty$, $\|\boldsymbol{\sigma}_j^2\|_\infty \leq 1$, and $n \leq n_j$ for $j = 1, 2, \ldots, k$. For any $j \in \{1, 2, \ldots, k\}$, we have that the ESS in (12) can be taken so that it satisfies

$$n_j^{\rightleftharpoons k} \geq n \left( 1 + \frac{(1+\alpha)^{2j-1} - 1 - \alpha}{\alpha(1+\alpha)^{2j-1} + \alpha} + \frac{(1+\alpha)^{2(k-j)+1} - 1 - \alpha}{\alpha(1+\alpha)^{2(k-j)+1} + \alpha} \right) \text{ with } \alpha = \frac{2}{\sqrt{1 + \frac{4}{nd^2}} - 1}. \tag{13}$$

In particular, for $j \geq 2$, we have that

$$n_j^{\rightleftharpoons k} \geq n_j^{\rightarrow} + n \frac{j(k-j)}{j + 2(k-j)} \geq n \left( 1 + \frac{j-1}{3} + \frac{j(k-j)}{j + 2(k-j)} \right) \quad \text{if} \quad nd^2 < \frac{1}{j^2}$$

$$n_j^{\rightleftharpoons k} \geq n_j^{\rightarrow} + \frac{1}{5}\sqrt{\frac{n}{d^2}} \quad \geq n \left( 1 + \frac{2}{5\sqrt{nd^2}} \right) \quad \text{if} \quad \frac{1}{j^2} \leq nd^2 < 1$$

$$n_j^{\rightleftharpoons k} \geq n_j^{\rightarrow} + \frac{1}{3d^2} \quad \geq n \left( 1 + \frac{2}{3nd^2} \right) \quad \text{if} \quad nd^2 \geq 1.$$

*Proof.* See Appendix F. □

The above theorem characterizes the increase in ESS provided by forward and backward learning in terms of the tasks' expected quadratic change. Such increase grows monotonically with the number of preceding tasks $j$ and with the number of succeeding tasks $k - j$ as shown in (13). In addition, it becomes proportional to the total number of tasks $k$ when the expected quadratic change is smaller than $1/(j^2 n)$ and $j \geq k/2$. Figure 3 further illustrates the increase in ESS with respect to the sample size $(n_j^{\rightleftharpoons k}/n)$ due to forward and backward learning in comparison with forward learning. Such figure displays the three intervals that are discussed in Theorems 2 and 4. In particular, the ESS significantly increases when $nd^2$ decreases between 1 and $1/j^2$. Note also that in most of the situations, the benefits of backward learning are achieved using only $b = k - j = 3$ backward steps.

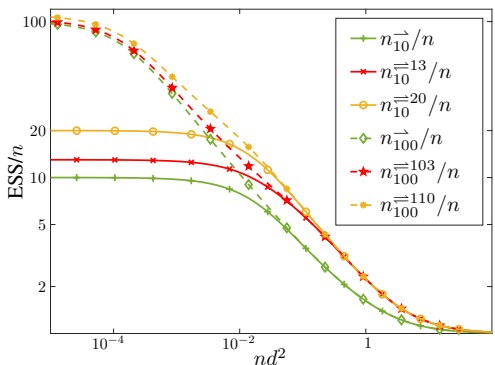

Figure 3: ESS increase provided by forward and backward learning.

## 5 NUMERICAL RESULTS

This section first compares the classification performance of LMRCs with state-of-the-art techniques using multiple datasets, then we show the performance improvement of the presented LMRCs due forward and backward learning. In the supplementary materials, we provide the code for LMRCs together with additional implementation details and numerical results in Appendix G.

The proposed method is evaluated using 6 public datasets: "Yearbook" (Ginosar et al., 2015), "ImageNet noise" (Mai et al., 2022), "UTKFaces" (Zhang et al., 2017), "Rotated MNIST" (Jin et al., 2021), "DomainNet" (Peng et al., 2019), and "CLEAR" (Lin et al., 2021). These datasets are composed by time-dependent tasks (images with characteristics/quality/realism that change over time). The last two datasets are multi-class problems and the rest are binary (see further details in Appendix G). For all methods, instances are represented by pixel values in "Rotated MNIST" dataset, and by the last layer of the ResNet18 pre-trained network (He et al., 2016) in the remaining datasets. The proposed LMRC method is compared with 4 lifelong learning techniques: gradient episodic memory (GEM) (Lopez-Paz & Ranzato, 2017), meta-experience replay (MER) (Riemer et al., 2018), efficient lifelong learning algorithm (ELLA) (Ruvolo & Eaton, 2013), and elastic weight consolidation (EWC) (Kirkpatrick et al., 2017). The hyper-parameters in these methods are set to the default values provided by the authors. LMRCs are implemented using $b = 3$

Table 1: Classification error and standard deviation of the proposed LMRC method in comparison with the state-of-the-art techniques.

| Dataset | Yearbook | | ImageNet noise | | DomainNet | | UTKFaces | | Rotated MNIST | | CLEAR | |
|---|---|---|---|---|---|---|---|---|---|---|---|---|
| $n$ | 10 | 100 | 10 | 100 | 10 | 100 | 10 | 100 | 10 | 100 | 10 | 100 |
| GEM | $.18 \pm .03$ | $.17 \pm .03$ | $.39 \pm .08$ | $.13 \pm .07$ | $.69 \pm .05$ | $.53 \pm .10$ | $.12 \pm .00$ | $.12 \pm .00$ | $\mathbf{.36} \pm .06$ | $.28 \pm .02$ | $.57 \pm .10$ | $.09 \pm .02$ |
| MER | $.16 \pm .03$ | $.10 \pm .01$ | $.17 \pm .03$ | $.10 \pm .01$ | $.38 \pm .04$ | $\mathbf{.26} \pm .04$ | $.17 \pm .09$ | $.11 \pm .01$ | $.37 \pm .09$ | $.45 \pm .10$ | $.10 \pm .03$ | $\mathbf{.05} \pm .02$ |
| ELLA | $.45 \pm .09$ | $.43 \pm .10$ | $.48 \pm .05$ | $.47 \pm .04$ | $.67 \pm .05$ | $.67 \pm .05$ | $.19 \pm .12$ | $.17 \pm .11$ | $.48 \pm .05$ | $.47 \pm .05$ | $.61 \pm .06$ | $.60 \pm .05$ |
| EWC | $.47 \pm .05$ | $.27 \pm .06$ | $.47 \pm .04$ | $.46 \pm .06$ | $.75 \pm .04$ | $.74 \pm .05$ | $.12 \pm .00$ | $.12 \pm .00$ | $.48 \pm .01$ | $.40 \pm .01$ | $.65 \pm .03$ | $.62 \pm .04$ |
| LMRC | $\mathbf{.13} \pm .04$ | $\mathbf{.08} \pm .02$ | $\mathbf{.15} \pm .03$ | $\mathbf{.09} \pm .01$ | $\mathbf{.34} \pm .06$ | $.28 \pm .01$ | $\mathbf{.10} \pm .01$ | $\mathbf{.10} \pm .00$ | $\mathbf{.36} \pm .01$ | $\mathbf{.21} \pm .00$ | $\mathbf{.09} \pm .03$ | $\mathbf{.05} \pm .02$ |

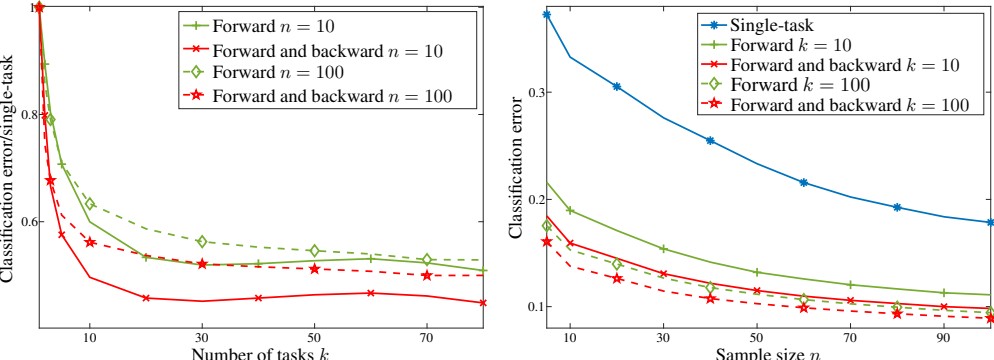

(a) Classification error per number of tasks in "Yearbook" dataset.

(b) Classification error per sample size in "Yearbook" dataset.

Figure 4: Forward and backward learning can sharply boost performance and ESS as tasks arrive.

backward steps and the expected quadratic change $d_j^2$ is estimated using $W = 2$ in (7). We use the same hyper-parameters for all the results in this section for fair comparison with the state-of-the-art and to show that the techniques presented do not require a careful fine-tuning. In Appendix G, among other additional results, we study the change in classification error and processing time achieved by varying the number $b$ of backward steps.

In the first set of numerical results, we compare the performance of the proposed LMRCs with the state-of-the-art techniques for $n = 10$ and $n = 100$ samples per task. These numerical results are obtained computing the average classification error over all the tasks in 50 random instantiations of data samples. As can be observed in Table 1, LMRCs can significantly improve performance in time-dependent tasks with respect to the state-of-the-art.

In the second set of numerical results, we analyze the contribution of forward and backward learning to the final performance of LMRCs. In particular, we show the relationship among classification error, number of tasks, and sample size for single-task, forward, and forward and backward learning. These numerical results are obtained averaging, for each number of tasks and sample size, the classification errors achieved with 10 random instantiations of data samples in "Yearbook" dataset (see Appendix G for further details). Figure 4a shows the classification error of LMRC method divided by the classification error of single-task learning for different number of tasks with $n = 10$ and $n = 100$ sample sizes. Such figure shows that forward and backward learning can significantly improve performance as tasks arrive. In addition, Figure 4b shows the classification error of LMRC method for different sample sizes with $k = 10$ and $k = 100$ tasks. Such figure shows that forward and backward learning for $k = 100$ tasks using $n = 10$ samples achieves significantly better results than single-task learning using $n = 100$ samples. In particular, the methods proposed can effectively exploit backward learning that results in enhanced classification error in all the experimental results.

## 6 CONCLUSION

The paper proposes LMRCs that effectively perform forward and backward learning and account for time-dependent tasks. LMRCs carefully avoid the repeated usage of the same information from the ever-increasing sequence of succeeding tasks. In addition, the paper analytically characterizes the increase in ESS achieved by the proposed forward and backward learning techniques in terms of the tasks' expected quadratic change and number of tasks. The numerical results assess the performance improvement of LMRC methodology with respect to the state-of-the-art using multiple datasets, sample sizes, and number of tasks. The proposed methodology for lifelong learning with time-dependent tasks can lead to techniques that further approach the humans' ability to learn from few examples and to continuously improve on tasks that arrive over time.

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

## A  DERIVATION OF RECURSIONS (5) AND (6) FOR FORWARD LEARNING AND RECURSIONS (10) AND (11) FOR FORWARD AND BACKWARD LEARNING

This section shows how recursions in (5), (6) and recursions in (10), (11) are obtained using those for filtering and smoothing in linear dynamical systems.

The mean vectors evolve over time steps through the linear dynamical system

$$\boldsymbol{\tau}_j^\infty = \boldsymbol{\tau}_{j-1}^\infty + \boldsymbol{w}_j \tag{14}$$

where, as described in Section 3.1, vectors $\boldsymbol{w}_j$ for $j \in \{2, 3, \ldots, k\}$ are independent and zero-mean because $\mathrm{p}_j - \mathrm{p}_{j-1}$ are independent and zero-mean. In addition, each state variable $\boldsymbol{\tau}_j^\infty$ is observed at each step $j$ through $\boldsymbol{\tau}_j$ that is the sample average of i.i.d. samples from $\mathrm{p}_j$, so that we have

$$\boldsymbol{\tau}_j = \boldsymbol{\tau}_j^\infty + \boldsymbol{v}_j \tag{15}$$

where $\boldsymbol{v}_j$ for $j \in \{1, 2, \ldots, k\}$ are independent and zero mean, and independent of $\boldsymbol{w}_j$ for $j \in \{1, 2, \ldots, k\}$. Therefore, equations (14) and (15) above describe a linear dynamical system (state-space model with white noise processes) (Bishop, 2006; Anderson & Moore, 1979). For such systems, the Kalman filter recursions provide the unbiased linear estimator with minimum MSE based on samples corresponding to preceding steps $D_1, D_2, \ldots, D_j$, and fixed-lag smoother recursions provide the unbiased linear estimator with minimum MSE based on samples corresponding to preceding and succeeding steps $D_1, D_2, \ldots, D_k$ (Bishop, 2006; Anderson & Moore, 1979). Then, equations (5), (6) and equations (10), (11) are obtained after some algebra from the Kalman filter recursions and fixed-lag smoother recursions, respectively.

## B  PROOF OF THEOREM 1

*Proof.* To obtain bound in (8) we first prove that the mean vector estimate and the MSE vector given by (5) and (6), respectively, satisfy

$$\mathbb{P}\left\{ |\tau_j^{\infty (i)} - \tau_j^{\rightarrow (i)}| \leq \kappa \sqrt{2 s_j^{\rightarrow (i)} \log\left(\frac{2m}{\delta}\right)} \right\} \geq (1 - \delta) \tag{16}$$

for any component $i = 1, 2, \ldots, m$. Then, we prove that $\|\sqrt{\boldsymbol{s}_j^{\rightarrow}}\|_\infty \leq M / \sqrt{n_j^{\rightarrow}}$ for $j \in \{1, 2, \ldots, k\}$, where the ESSs satisfy $n_1^{\rightarrow} = n_1$ and $n_j^{\rightarrow} \geq n_j + n_{j-1}^{\rightarrow} \frac{\|\boldsymbol{\sigma}_j^2\|_\infty}{\|\boldsymbol{\sigma}_j^2\|_\infty + n_{j-1}^{\rightarrow} \|\boldsymbol{d}_j^2\|_\infty}$ for $j \geq 2$.

To obtain inequality (16), we prove by induction that each component $i = 1, 2, \ldots, m$ of the error in the mean vector estimate $z_j^{\rightarrow (i)} = \tau_j^{\infty (i)} - \tau_j^{\rightarrow (i)}$ is sub-Gaussian with parameter $\eta_j^{\rightarrow (i)} \leq \kappa \sqrt{s_j^{\rightarrow (i)}}$. Firstly, for $j = 1$, we have that

$$z_1^{\rightarrow (i)} = \tau_1^{\infty (i)} - \tau_1^{\rightarrow (i)} = \tau_1^{\infty (i)} - \tau_1^{(i)}.$$

Since the bounded random variable $\Phi_1^{(i)}$ is sub-Gaussian with parameter $\sigma(\Phi_1^{(i)})$, then the error in the mean vector estimate $z_1^{\rightarrow (i)}$ is sub-Gaussian with parameter that satisfies

$$\left( \eta_1^{\rightarrow (i)} \right)^2 = \frac{\sigma\left(\Phi_1^{(i)}\right)^2}{n_1} \leq \frac{\kappa^2 \sigma_1^{2(i)}}{n_1} = \kappa^2 s_1^{(i)}.$$

If $z_{j-1}^{\rightarrow (i)} = \tau_{j-1}^{\infty (i)} - \tau_{j-1}^{\rightarrow (i)}$ is sub-Gaussian with parameter $\eta_{j-1}^{\rightarrow (i)} \leq \kappa \sqrt{s_{j-1}^{\rightarrow (i)}}$ for any $i = 1, 2, \ldots, m$, then using the recursions (5) and (6) we have that

$$z_j^{\rightarrow (i)} = \tau_j^{\infty (i)} - \tau_j^{\rightarrow (i)} = \tau_{j-1}^{\infty (i)} + w_j^{(i)} - \tau_j^{(i)} - \frac{s_j^{(i)}}{s_{j-1}^{\rightarrow (i)} + s_j^{(i)} + d_j^{2(i)}} \left( \tau_{j-1}^{\rightarrow (i)} - \tau_j^{(i)} \right)$$

$$= \tau_{j-1}^{\infty (i)} + w_j^{(i)} - \tau_{j-1}^{\rightarrow (i)} + \left( 1 - \frac{s_j^{(i)}}{s_{j-1}^{\rightarrow (i)} + s_j^{(i)} + d_j^{2(i)}} \right) \left( \tau_{j-1}^{\rightarrow (i)} - \tau_j^{(i)} \right)$$

$$= \tau_{j-1}^{\infty (i)} + w_j^{(i)} - \tau_{j-1}^{\rightarrow (i)} - \frac{s_j^{\rightarrow (i)}}{s_j^{(i)}} \left( \tau_j^{(i)} - \tau_{j-1}^{\rightarrow (i)} \right)$$

since $\boldsymbol{w}_j = \boldsymbol{\tau}_j^\infty - \boldsymbol{\tau}_{j-1}^\infty$. If $\boldsymbol{v}_j = \boldsymbol{\tau}_j - \boldsymbol{\tau}_j^\infty$, the error in the mean vector estimate is given by

$$
\begin{aligned}
z_j^{\rightarrow(i)} &= \tau_{j-1}^{\infty\,(i)} + w_j^{(i)} - \tau_{j-1}^{\rightarrow(i)} - \frac{s_j^{\rightarrow(i)}}{s_j^{(i)}}\left(\tau_j^{\infty\,(i)} + v_j^{(i)} - \tau_{j-1}^{\rightarrow(i)}\right) \\
&= \tau_{j-1}^{\infty\,(i)} + w_j^{(i)} - \tau_{j-1}^{\rightarrow(i)} - \frac{s_j^{\rightarrow(i)}}{s_j^{(i)}}\left(\tau_{j-1}^{\infty\,(i)} + w_j^{(i)} + v_j^{(i)} - \tau_{j-1}^{\rightarrow(i)}\right) \\
&= \left(1 - \frac{s_j^{\rightarrow(i)}}{s_j^{(i)}}\right) z_{j-1}^{\rightarrow(i)} + \left(1 - \frac{s_j^{\rightarrow(i)}}{s_j^{(i)}}\right)\left(w_j^{(i)}\right) - \frac{s_j^{\rightarrow(i)}}{s_j^{(i)}} v_j^{(i)}
\end{aligned}
$$

where $w_j^{(i)}$ and $v_j^{(i)}$ are sub-Gaussian with parameter $\sigma(w_j^{(i)})$ and $\sigma\left(\Phi_j^{(i)}\right)/\sqrt{n_j}$, respectively. Therefore, we have that $z_j^{\rightarrow(i)}$ is sub-Gaussian with parameter $\eta_j^{\rightarrow(i)}$ that satisfies

$$
\left(\eta_j^{\rightarrow(i)}\right)^2 = \left(1 - \frac{s_j^{\rightarrow(i)}}{s_j^{(i)}}\right)^2 \left(\eta_{j-1}^{\rightarrow(i)}\right)^2 + \left(1 - \frac{s_j^{\rightarrow(i)}}{s_j^{(i)}}\right)^2 \sigma\left(w_j^{(i)}\right)^2 + \left(\frac{s_j^{\rightarrow(i)}}{s_j^{(i)}}\right)^2 \frac{\sigma\left(\Phi_j^{(i)}\right)^2}{n_j}
$$

since $\boldsymbol{z}_{j-1}^{\rightarrow}$, $\boldsymbol{w}_j$, and $\boldsymbol{v}_j$ are independent. Using that $\eta_{j-1}^{\rightarrow(i)} \le \kappa\sqrt{s_{j-1}^{\rightarrow(i)}}$ and the definition of $\kappa$, we have that

$$
\begin{aligned}
\left(\eta_j^{\rightarrow(i)}\right)^2 &\le \left(1 - \frac{s_j^{\rightarrow(i)}}{s_j^{(i)}}\right)^2 \kappa^2 s_{j-1}^{\rightarrow(i)} + \left(1 - \frac{s_j^{\rightarrow(i)}}{s_j^{(i)}}\right)^2 \kappa^2 d_j^{2\,(i)} + \left(\frac{s_j^{\rightarrow(i)}}{s_j^{(i)}}\right)^2 \kappa^2 \frac{\sigma_j^{2\,(i)}}{n_j} \\
&\le \left(1 - \frac{s_j^{\rightarrow(i)}}{s_j^{(i)}}\right)^2 \kappa^2 \left(\left(\frac{1}{s_j^{\rightarrow(i)}} - \frac{1}{s_j^{(i)}}\right)^{-1} + d_j^{2\,(i)}\right) + \frac{\left(s_j^{\rightarrow(i)}\right)^2}{s_j^{(i)}}\kappa^2 \qquad (17) \\
&= \left(1 - \frac{s_j^{\rightarrow(i)}}{s_j^{(i)}}\right) \kappa^2 s_j^{\rightarrow(i)} + \kappa^2 \frac{\left(s_j^{\rightarrow(i)}\right)^2}{s_j^{(i)}}
\end{aligned}
$$

where (17) is obtained using (6).

The inequality in (16) is obtained using the union bound together with the Chernoff bound (concentration inequality) (Wainwright, 2019) for the random variables $z_j^{\rightarrow(i)}$ that are sub-Gaussian with parameter $\eta_j^{\rightarrow(i)}$.

Now, we prove by induction that, for any $j$, $\|\sqrt{\boldsymbol{s}_j^{\rightarrow}}\|_\infty \le M/\sqrt{n_j^{\rightarrow}}$ where the ESSs satisfy $n_1^{\rightarrow} = n_1$ and $n_j^{\rightarrow} \ge n_j + n_{j-1}^{\rightarrow}\frac{\|\boldsymbol{\sigma}_j^2\|_\infty}{\|\boldsymbol{\sigma}_j^2\|_\infty + n_{j-1}^{\rightarrow}\|\boldsymbol{d}_j^2\|_\infty}$ for $j \ge 2$. For $j = 1$, using the definition of $\boldsymbol{s}_j^{\rightarrow}$ in equation (6), we have that for any component $i$

$$
\left(s_1^{\rightarrow(i)}\right)^{-1} = \left(s_1^{(i)}\right)^{-1} = \frac{n_1}{\sigma_1^{2\,(i)}} \ge \frac{n_1}{M^2}.
$$

Then, vector $\boldsymbol{s}_1^{\rightarrow}$ satisfies

$$
\|\sqrt{\boldsymbol{s}_1^{\rightarrow}}\|_\infty \le \frac{M}{\sqrt{n_1}} = \frac{M}{\sqrt{n_1^{\rightarrow}}}.
$$

If $\|\sqrt{\boldsymbol{s}_{j-1}^{\rightarrow}}\|_\infty \le M/\sqrt{n_{j-1}^{\rightarrow}}$, then we have that for any component $i$

$$
\begin{aligned}
\left(s_j^{\rightarrow(i)}\right)^{-1} &= \frac{1}{s_j^{(i)}} + \frac{1}{s_{j-1}^{\rightarrow(i)} + d_j^{2\,(i)}} \ge \frac{1}{s_j^{(i)}} + \frac{1}{\frac{M^2}{n_{j-1}^{\rightarrow}} + d_j^{2\,(i)}} \ge \frac{1}{M^2}\left(n_j + \frac{1}{\frac{1}{n_{j-1}^{\rightarrow}} + \frac{d_j^{2\,(i)}}{M^2}}\right) \\
&\ge \frac{1}{M^2}\left(n_j + \frac{1}{\frac{1}{n_{j-1}^{\rightarrow}} + \frac{\|\boldsymbol{d}_j^2\|_\infty}{\|\boldsymbol{\sigma}_j^2\|_\infty}}\right)
\end{aligned}
$$

by using the recursion (6) and the induction hypothesis. Then, vector $\vec{s_j}$ satisfies

$$\left\| \sqrt{\vec{s_j}} \right\|_\infty \leq \frac{M}{\sqrt{n_j + \vec{n}_{j-1} \frac{\|\boldsymbol{\sigma}_j^2\|_\infty}{\|\boldsymbol{\sigma}_j^2\|_\infty + \vec{n}_{j-1}\|\boldsymbol{d}_j^2\|_\infty}}}. \tag{18}$$

The inequality in (8) is obtained because the minimax risk is bounded by the smallest minimax risk as shown in (Mazuelas et al., 2020; 2022a;b) so that

$$R(\vec{\mathcal{U}_j}) \leq R_j^\infty + \left( \|\boldsymbol{\tau}_j^\infty - \vec{\boldsymbol{\tau}_j}\|_\infty + \|\vec{\boldsymbol{\lambda}_j}\|_\infty \right) \|\boldsymbol{\mu}_j^\infty\|_1$$

that leads to (8) using (16), (18), and the fact that $1 \leq \sqrt{2 \log\left(\frac{2m}{\delta}\right)}$. $\qquad\square$

## C    PROOF OF THEOREM 2

*Proof.* To obtain bound in (9), we proceed by induction. For $j = 1$, using the expression for the ESS in (8), we have that

$$\vec{n_1} = n_1 \geq n.$$

If (9) holds for the $(j-1)$-task, then for the $j$-th task, we have that

$$\vec{n_j} \geq n_j + \vec{n}_{j-1} \frac{\|\boldsymbol{\sigma}_j^2\|_\infty}{\|\boldsymbol{\sigma}_j^2\|_\infty + \vec{n}_{j-1}\|\boldsymbol{d}_j^2\|_\infty} \geq n + \vec{n}_{j-1} \frac{1}{1 + \vec{n}_{j-1}d^2} = n \left( 1 + \frac{1}{\frac{n}{\vec{n}_{j-1}} + nd^2} \right)$$

where the second inequality is obtained because $n_j \geq n$, $\|\boldsymbol{\sigma}_j^2\|_\infty \leq 1$, and $\|\boldsymbol{d}_j^2\|_\infty \leq d^2$. Using that $\vec{n}_{j-1} \geq n \left( 1 + \frac{(1+\alpha)^{2j-3}-1-\alpha}{\alpha(1+\alpha)^{2j-3}+\alpha} \right)$, the ESS of the $j$-th task satisfies

$$\begin{aligned}
\vec{n_j} &\geq n \left( 1 + \frac{1}{\frac{n}{n\left(1+\frac{(1+\alpha)^{2j-3}-1-\alpha}{\alpha(1+\alpha)^{2j-3}+\alpha}\right)} + nd^2} \right) = n \left( 1 + \frac{1}{\frac{\alpha(1+\alpha)^{2j-3}+\alpha}{(1+\alpha)^{2j-2}-1} + nd^2} \right) \\
&= n \left( 1 + \frac{1}{\frac{\alpha(1+\alpha)^{2j-3}+\alpha}{(1+\alpha)^{2j-2}-1} + \frac{\alpha^2}{\alpha+1}} \right) \tag{19} \\
&= n \left( 1 + \frac{(1+\alpha)^{2j-1}-1-\alpha}{\alpha(1+\alpha)^{2j-2}+\alpha(\alpha+1)+\alpha^2(1+\alpha)^{2j-2}-\alpha^2} \right)
\end{aligned}$$

where (19) is obtained because $nd^2 = \frac{\alpha^2}{\alpha+1}$ since $\alpha = \frac{nd^2}{2}\left(\sqrt{1+\frac{4}{nd^2}}+1\right)$.

Now, we obtain bounds for the ESS depending on the value of $nd^2$. In the following, the constant $\phi$ represents the golden ratio $\phi = 1.618\ldots$.

1. If $nd^2 < \frac{1}{j^2} \Rightarrow \sqrt{nd^2} \leq \alpha \leq \sqrt{nd^2}\phi \leq \frac{\phi}{j} \leq 1$ similarly as in the previous case, then we have that $\vec{n_j}$ satisfies

$$\vec{n_j} \geq n \left( 1 + \frac{1}{\alpha} \frac{\alpha(2j-2)}{2+\alpha(2j-1)} \right) = n \left( 1 + \frac{2j-2}{2+\alpha(2j-1)} \right)$$

where the first inequality follows because $(1+\alpha)^{2j-2} \geq 1 + \alpha(2j-2)$. Using $\alpha \leq \frac{\phi}{j}$, we have that

$$\vec{n_j} \geq n \left( 1 + \frac{2j-2}{2+\frac{\phi}{j}(2j-1)} \right) \geq n \left( 1 + \frac{2j-2}{2+2\phi-\frac{\phi}{j}} \right) \geq n \left( 1 + \frac{j-1}{1+\phi} \right).$$

2. If $\frac{1}{j^2} \leq nd^2 < 1 \Rightarrow \frac{1}{j} < \sqrt{nd^2} < \alpha < \sqrt{nd^2}\phi$ because $\alpha = \frac{nd^2}{2}\left(\sqrt{1+\frac{4}{nd^2}}+1\right) = \sqrt{nd^2}\frac{\sqrt{nd^2+4}+\sqrt{nd^2}}{2}$, then we have that $n_j^{\rightarrow}$ satisfies

$$n_j^{\rightarrow} \geq n\left(1 + \frac{1}{\sqrt{nd^2}}\frac{1}{\phi}\frac{(1+\alpha)^{2j-2}-1}{(1+\alpha)^{2j-2}+1}\right) \geq n\left(1 + \frac{1}{\sqrt{nd^2}}\frac{1}{\phi}\frac{(1+\frac{1}{j})^{2j-2}-1}{(1+\frac{1}{j})^{2j-2}+1}\right)$$

where the first inequality follows because $\alpha < \sqrt{nd^2}\phi$ and the second inequality follows because the expression is monotonically increasing for $\alpha$ and $\frac{1}{j} < \alpha$. Since $(1+\frac{1}{j})^{2j-2} \geq 1 + \frac{2j-2}{j}$, we have that

$$n_j^{\rightarrow} \geq n\left(1 + \frac{1}{\sqrt{nd^2}}\frac{1}{\phi}\frac{\frac{2j-2}{j}}{2+\frac{2j-2}{j}}\right) \geq n\left(1 + \frac{1}{\sqrt{nd^2}}\frac{1}{\phi}\frac{1}{3}\right)$$

because $j \geq 2$.

3. If $nd^2 \geq 1 \Rightarrow 1 \leq nd^2 \leq \alpha \leq nd^2\phi$ because $\alpha = \frac{nd^2}{2}\left(\sqrt{1+\frac{4}{nd^2}}+1\right)$, then we have that $n_j^{\rightarrow}$ satisfies

$$n_j^{\rightarrow} \geq n\left(1 + \frac{1}{nd^2\phi}\frac{(1+\alpha)^{2j-1}-1-\alpha}{(1+\alpha)^{2j-1}+1}\right) \geq n\left(1 + \frac{1}{nd^2\phi}\frac{(1+\alpha)^{2j-2}-1}{(1+\alpha)^{2j-2}+1}\right)$$

where the first inequality follows because $\alpha \leq nd^2\phi$ and the second inequality follows multiplying and dividing by $1+\alpha$ and because $1/(1+\alpha) < 1$. Since the above expression is monotonically increasing for $\alpha$ and $\alpha \geq 1$, we have that

$$n_j^{\rightarrow} \geq n\left(1 + \frac{1}{nd^2\phi}\frac{2^{2j-2}-1}{2^{2j-2}+1}\right) \geq n\left(1 + \frac{1}{nd^2\phi}\frac{3}{5}\right)$$

because $j \geq 2$.

$\square$

## D  MORE EFFICIENT RECURSIONS FOR FORWARD AND BACKWARD LEARNING

The Rauch-Tung-Striebel smoother recursions (Bishop, 2006; Anderson & Moore, 1979) allow to obtain forward and backward mean and MSE vectors directly from those vectors for the succeeding task. Specifically, for each $j$-th task, the mean vector $\boldsymbol{\tau}_j^{\rightleftharpoons k}$ together with the MSE vector $\boldsymbol{s}_j^{\rightleftharpoons k}$ can be obtained using those vectors for the succeeding task $\boldsymbol{\tau}_{j+1}^{\rightleftharpoons k}, \boldsymbol{s}_{j+1}^{\rightleftharpoons k}$ as

$$\boldsymbol{\tau}_j^{\rightleftharpoons k} = \boldsymbol{\tau}_j^{\rightarrow} + \frac{\boldsymbol{s}_j^{\rightarrow}}{\boldsymbol{s}_j^{\rightarrow} + \boldsymbol{d}_{j+1}^2}\left(\boldsymbol{\tau}_{j+1}^{\rightleftharpoons k} - \boldsymbol{\tau}_j^{\rightarrow}\right)$$

$$\boldsymbol{s}_j^{\rightleftharpoons k} = \left(\frac{1}{\boldsymbol{s}_j^{\rightarrow}} + \left(\boldsymbol{d}_{j+1}^2 + \left(\frac{1}{\boldsymbol{s}_{j+1}^{\rightleftharpoons k}} - \frac{1}{\boldsymbol{s}_j^{\rightarrow} + \boldsymbol{d}_{j+1}^2}\right)^{-1}\right)^{-1}\right)^{-1}.$$

The above recursions provide the same mean vector estimate as the recursions (10) and (11) in the paper since they are obtained using the Rauch-Tung-Striebel smoother recursions instead of fixed-lag smoother recursions (Bishop, 2006; Anderson & Moore, 1979).

## E  PROOF OF THEOREM 3

*Proof.* To obtain bound in (12) we first prove that the mean vector estimate and the MSE vector given by (10) and (11), respectively, satisfy

$$\mathbb{P}\left\{|\tau_k^{\infty\,(i)} - \tau_j^{\rightleftharpoons k\,(i)}| \leq \kappa\sqrt{2s_j^{\rightleftharpoons k\,(i)}\log\left(\frac{2m}{\delta}\right)}\right\} \geq (1-\delta) \tag{20}$$

for any component $i = 1, 2, \ldots, m$. Then, we prove that $\|s_j^{\rightleftharpoons k}\|_\infty \leq M/\sqrt{n_j^{\rightleftharpoons k}}$ for $j \in \{1, 2, \ldots, k\}$, where the ESSs satisfy $n_k^{\rightleftharpoons k} = n_k^{\rightarrow}$ and $n_j^{\rightleftharpoons k} \geq n_j^{\rightarrow} + n_{j+1}^{\leftharpoondown k} \frac{\|\boldsymbol{\sigma}_j^2\|_\infty}{\|\boldsymbol{\sigma}_j^2\|_\infty + n_{j+1}^{\leftharpoondown k}\|\boldsymbol{d}_{j+1}^2\|_\infty}$ for $j \geq 2$.

To obtain inequality (20), we prove that each component $i = 1, 2, \ldots, m$ of the error in the mean vector estimate $z_j^{\rightleftharpoons k\,(i)} = \tau_j^{\infty\,(i)} - \tau_j^{\rightleftharpoons k\,(i)}$ is sub-Gaussian with parameter $\eta_j^{\rightleftharpoons k\,(i)} \leq \kappa\sqrt{s_j^{\rightleftharpoons k\,(i)}}$. Analogously to the proof of Theorem 1, it is proven that each component in the error of the backward mean vector $\boldsymbol{\tau}_{j+1}^{\leftharpoondown k}$ is sub-Gaussian with parameters satisfying $\boldsymbol{\eta}_{j+1}^{\leftharpoondown k} \preceq \kappa\sqrt{\boldsymbol{s}_{j+1}^{\leftharpoondown k}}$. The error in the forward and backward mean vector estimate is given by

$$z_j^{\rightleftharpoons k\,(i)} = \tau_j^{\infty\,(i)} - \tau_j^{\rightleftharpoons k\,(i)} = \tau_j^{\infty\,(i)} - \tau_j^{\rightarrow\,(i)} - \frac{s_j^{\rightarrow\,(i)}}{s_j^{\rightarrow\,(i)} + s_{j+1}^{\leftharpoondown k\,(i)} + d_{j+1}^{2\,(i)}}\left(\tau_{j+1}^{\leftharpoondown k\,(i)} - \tau_j^{\rightarrow\,(i)}\right)$$

where the second equality is obtained using the recursion for $\tau_j^{\rightleftharpoons k\,(i)}$ in (10). Adding and subtracting $\frac{s_j^{\rightarrow\,(i)}}{s_j^{\rightarrow\,(i)} + s_{j+1}^{\leftharpoondown k\,(i)} + d_{j+1}^{2\,(i)}}\tau_{j+1}^{\infty\,(i)}$, we have that

$$z_j^{\rightleftharpoons k\,(i)} = z_j^{\rightarrow\,(i)} - \frac{s_j^{\rightarrow\,(i)}}{s_j^{\rightarrow\,(i)} + s_{j+1}^{\leftharpoondown k\,(i)} + d_{j+1}^{2\,(i)}}\left(\tau_{j+1}^{\infty\,(i)} - \tau_{j+1}^{\infty\,(i)} + \tau_{j+1}^{\leftharpoondown k\,(i)} - \tau_j^{\rightarrow\,(i)}\right)$$

$$= z_j^{\rightarrow\,(i)} - \frac{s_j^{\rightarrow\,(i)}}{s_j^{\rightarrow\,(i)} + s_{j+1}^{\leftharpoondown k\,(i)} + d_{j+1}^{2\,(i)}}\left(\tau_j^{\infty\,(i)} + w_{j+1}^{(i)} - z_{j+1}^{\leftharpoondown k\,(i)} - \tau_j^{\rightarrow\,(i)}\right)$$

since $\boldsymbol{w}_j = \boldsymbol{\tau}_j^{\infty} - \boldsymbol{\tau}_{j-1}^{\infty}$ and $z_j^{\rightarrow\,(i)} = \tau_j^{\infty\,(i)} - \tau_j^{\rightarrow\,(i)}$. Then, we have that

$$z_j^{\rightleftharpoons k\,(i)} = z_j^{\rightarrow\,(i)} - \frac{s_j^{\rightarrow\,(i)}}{s_j^{\rightarrow\,(i)} + s_{j+1}^{\leftharpoondown k\,(i)} + d_{j+1}^{2\,(i)}}\left(z_j^{\rightarrow\,(i)} + w_{j+1}^{(i)} - z_{j+1}^{\leftharpoondown k\,(i)}\right) \qquad (21)$$

$$= \left(1 - \frac{s_j^{\rightarrow\,(i)}}{s_j^{\rightarrow\,(i)} + s_{j+1}^{\leftharpoondown k\,(i)} + d_{j+1}^{2\,(i)}}\right)z_j^{\rightarrow\,(i)}$$

$$- \frac{s_j^{\rightarrow\,(i)}}{s_j^{\rightarrow\,(i)} + s_{j+1}^{\leftharpoondown k\,(i)} + d_{j+1}^{2\,(i)}}\left(w_{j+1}^{(i)} - z_{j+1}^{\leftharpoondown k\,(i)}\right)$$

where $z_j^{\rightarrow\,(i)}$, $z_{j+1}^{\leftharpoondown k\,(i)}$, and $w_{j+1}^{(i)}$ are sub-Gaussian with parameters $\eta_j^{\rightarrow\,(i)} \leq \kappa\sqrt{s_j^{\rightarrow\,(i)}}$, $\eta_{j+1}^{\leftharpoondown k\,(i)} \leq \kappa\sqrt{s_{j+1}^{\leftharpoondown k\,(i)}}$, and $\sigma(w_j^{(i)})$, respectively. Since $\boldsymbol{z}_j^{\rightarrow}$, $\boldsymbol{z}_{j+1}^{\leftharpoondown k}$, and $\boldsymbol{w}_{j+1}$ are independent, we have that $z_j^{\rightleftharpoons k\,(i)}$ given by (21) is sub-Gaussian with parameter that satisfies

$$\left(\eta_j^{\rightleftharpoons k\,(i)}\right)^2 = \left(1 - \frac{s_j^{\rightarrow\,(i)}}{s_j^{\rightarrow\,(i)} + s_{j+1}^{\leftharpoondown k\,(i)} + d_{j+1}^{2\,(i)}}\right)^2\left(\eta_j^{\rightarrow\,(i)}\right)^2$$

$$+ \left(\frac{s_j^{\rightarrow\,(i)}}{s_j^{\rightarrow\,(i)} + s_{j+1}^{\leftharpoondown k\,(i)} + d_{j+1}^{2\,(i)}}\right)^2\left(\sigma\left(w_j^{(i)}\right)^2 + \left(\eta_{j+1}^{\leftharpoondown k\,(i)}\right)^2\right)$$

$$\leq \left(1 - \frac{s_j^{\rightarrow\,(i)}}{s_j^{\rightarrow\,(i)} + s_{j+1}^{\leftharpoondown k\,(i)} + d_{j+1}^{2\,(i)}}\right)^2\kappa^2 s_j^{\rightarrow\,(i)}$$

$$+ \left(\frac{s_j^{\rightarrow\,(i)}}{s_j^{\rightarrow\,(i)} + s_{j+1}^{\leftharpoondown k\,(i)} + d_{j+1}^{2\,(i)}}\right)^2\kappa^2\left(d_{j+1}^{(i)} + s_{j+1}^{\leftharpoondown k\,(i)}\right)$$

Using (11) we have that the sub-Gaussian parameter satisfies

$$
\left(\eta_j^{\rightleftharpoons k\,(i)}\right)^2 \leq \left(1 - \frac{s_j^{\rightleftharpoons k\,(i)}}{s_{j+1}^{\leftharpoondown k\,(i)} + d_{j+1}^{2\,(i)}}\right)^2 \kappa^2 \left(\frac{1}{s_j^{\rightleftharpoons k\,(i)}} - \frac{1}{s_{j+1}^{\leftharpoondown k^2\,(i)} + d_{j+1}^{2\,(i)}}\right)^{-1}
$$
$$
+ \frac{\left(s_j^{\rightleftharpoons k\,(i)}\right)^2}{s_{j+1}^{\leftharpoondown k\,(i)} + d_{j+1}^{2\,(i)}} \kappa^2
$$
$$
= \left(\frac{s_{j+1}^{\leftharpoondown k\,(i)} + d_{j+1}^{2\,(i)} - s_j^{\rightleftharpoons k\,(i)}}{s_{j+1}^{\leftharpoondown k\,(i)} + d_{j+1}^{2\,(i)}}\right) \kappa^2 s_j^{\rightleftharpoons k\,(i)} + \frac{\left(s_j^{\rightleftharpoons k\,(i)}\right)^2}{s_{j+1}^{\leftharpoondown k\,(i)} + d_{j+1}^{2\,(i)}} \kappa^2 = \kappa^2 s_j^{\rightleftharpoons k\,(i)}.
$$

The inequality in (20) is obtained using the union bound together with the Chernoff bound (concentration inequality) (Wainwright, 2019) for the random variables $z_j^{\rightleftharpoons k\,(i)}$ that are sub-Gaussian with parameter $\eta_j^{\rightleftharpoons k\,(i)}$.

Now, we prove that, for any $j$, $\|\sqrt{s_j^{\rightleftharpoons k}}\| \leq M/\sqrt{n_j^{\rightleftharpoons k}}$ where the ESSs satisfy $n_k^{\rightleftharpoons k} = n_k^{\rightarrow}$ and $n_j^{\rightleftharpoons k} \geq n_j^{\rightarrow} + n_{j+1}^{\leftharpoondown k} \frac{\|\boldsymbol{\sigma}_j^2\|_\infty}{\|\boldsymbol{\sigma}_j^2\|_\infty + n_{j+1}^{\leftharpoondown k}\|\boldsymbol{d}_{j+1}^2\|_\infty}$ for $j \geq 2$. Analogously to the proof of Theorem 1, we prove that the backward MSE vector $s_{j+1}^{\leftharpoondown k}$ satisfies $\|\sqrt{s_{j+1}^{\leftharpoondown k}}\|_\infty \leq M/\sqrt{n_{j+1}^{\leftharpoondown k}}$. Then, using that $\|\sqrt{s_{j+1}^{\leftharpoondown k}}\|_\infty \leq M/\sqrt{n_{j+1}^{\leftharpoondown k}}$, we have that for every component $i$

$$
\left(s_j^{\rightleftharpoons k\,(i)}\right)^{-1} = \frac{1}{s_j^{\rightarrow\,(i)}} + \frac{1}{s_{j+1}^{\leftharpoondown k\,(i)} + d_{j+1}^{2\,(i)}} \geq \frac{n_j^{\rightarrow}}{\sigma_j^{2\,(i)}} + \frac{1}{\frac{M^2}{n_{j+1}^{\leftharpoondown k}} + d_{j+1}^{2\,(i)}}
$$
$$
\geq \frac{1}{M^2}\left(n_j^{\rightarrow} + \frac{1}{\frac{1}{n_{j+1}^{\leftharpoondown k}} + \frac{d_{j+1}^2}{M^2}}\right) \geq \frac{1}{M^2}\left(n_j^{\rightarrow} + \frac{1}{\frac{1}{n_{j+1}^{\leftharpoondown k}} + \frac{\|d_{j+1}^2\|_\infty}{\|\boldsymbol{\sigma}_j^2\|_\infty}}\right).
$$

Then, we obtain

$$
\|\sqrt{s_j^{\rightleftharpoons k}}\|_\infty \leq \frac{M}{\sqrt{n_j^{\rightarrow} + \frac{1}{\frac{1}{n_{j+1}^{\leftharpoondown k}} + \frac{\|\boldsymbol{d}_{j+1}^2\|_\infty}{\|\boldsymbol{\sigma}_j^2\|_\infty}}}}. \tag{22}
$$

The inequality in (12) is obtained because the minimax risk is bounded by the smallest minimax risk as shown in (Mazuelas et al., 2020; 2022a;b) so that

$$
R(\mathcal{U}_j^{\rightleftharpoons k}) \leq R_j^\infty + \left(\|\boldsymbol{\tau}_j^\infty - \boldsymbol{\tau}_j^{\rightleftharpoons k}\|_\infty + \|\boldsymbol{\lambda}_j^{\rightleftharpoons k}\|_\infty\right)\|\boldsymbol{\mu}_j^\infty\|_1
$$

that leads to (12) using (20), (22), and the fact that $1 \leq \sqrt{2\log\left(\frac{2m}{\delta}\right)}$. $\qquad\square$

## F  PROOF OF THEOREM 4

*Proof.* To obtain bound in (13), we use the ESS obtained with forward learning in Theorem 2 and obtained with backward learning. Analogously to the proof of Theorem 2, we prove that the ESS obtained at backward learning satisfies

$$
n_{j+1}^{\leftharpoondown k} \geq n_{j+1} + n_{j+2}^{\leftharpoondown k}\frac{\|\boldsymbol{\sigma}_{j+1}^2\|_\infty}{\|\boldsymbol{\sigma}_{j+1}^2\|_\infty + n_{j+2}^{\leftharpoondown k}\|\boldsymbol{d}_{j+2}^2\|_\infty} \geq n\left(1 + \frac{(1+\alpha)^{2(k-j)-1} - 1 - \alpha}{\alpha(1+\alpha)^{2(k-j)-1} + \alpha}\right).
$$

Therefore, the ESS obtained with forward an backward learning satisfies

$$n_j^{\rightleftharpoons k} \geq n_j^{\rightarrow} + n\left(1 + \frac{(1+\alpha)^{2(k-j)-1} - 1 - \alpha}{\alpha(1+\alpha)^{2(k-j)-1} + \alpha}\right)\left(1 + \frac{n\left(1 + \frac{(1+\alpha)^{2(k-j)-1} - 1 - \alpha}{\alpha(1+\alpha)^{2(k-j)-1} + \alpha}\right)}{nd^2}\right)^{-1}$$

$$= n_j^{\rightarrow} + n\frac{(1+\alpha)^{2(k-j)} - 1}{\alpha(1+\alpha)^{2(k-j)-1} + \alpha}\left(1 + \frac{\alpha^2}{\alpha+1}\left(1 + \frac{(1+\alpha)^{2(k-j)-1} - 1 - \alpha}{\alpha(1+\alpha)^{2(k-j)-1} + \alpha}\right)\right)^{-1}$$

where the second equality follows because $nd^2 = \frac{\alpha^2}{\alpha+1}$ since $\alpha = \frac{nd^2}{2}\left(\sqrt{1 + \frac{4}{nd^2}} + 1\right)$. Then, we have that

$$n_j^{\rightleftharpoons k} \geq n_j^{\rightarrow} + n\frac{(1+\alpha)^{2(k-j)} - 1}{\alpha(1+\alpha)^{2(k-j)-1} + \alpha}$$
$$\cdot \left(\frac{((1+\alpha)^{2(k-j)-1} + 1)(\alpha + 1 + \alpha^2) + \alpha((1+\alpha)^{2(k-j)-1} - 1 - \alpha)}{(\alpha+1)((1+\alpha)^{2(k-j)-1} + 1)}\right)^{-1}$$
$$\geq n_j^{\rightarrow} + n\frac{(1+\alpha)^{2(k-j)} - 1}{\alpha(1+\alpha)^{2(k-j)-1} + \alpha}\frac{(\alpha+1)((1+\alpha)^{2(k-j)-1} + 1)}{(1+\alpha)^{2(k-j)+1} + 1}.$$

Now, we obtain bounds for the ESS depending on the value value of $nd^2$. Such bounds are obtained similarly as in Theorem 2 and we also denote by $\phi$ the golden ratio $\phi = 1.618\ldots$.

1. If $nd^2 < \frac{1}{j^2} \Rightarrow \sqrt{nd^2} \leq \alpha \leq \sqrt{nd^2}\phi \leq \frac{\phi}{j} \leq 1$ similarly as in the previous case, then we have that $n_j^{\rightleftharpoons k}$ satisfies

$$n_j^{\rightleftharpoons k} \geq n_j^{\rightarrow} + n\frac{1}{\alpha}\frac{\alpha(2(k-j))}{2 + \alpha 2(k-j)} = n_j^{\rightarrow} + n\frac{k-j}{1 + \alpha(k-j)} \geq n_j^{\rightarrow} + n\frac{k-j}{1 + \frac{\phi}{j}(k-j)}$$

where the first inequality follows because $(1+\alpha)^{2(k-j)-1} \geq 1 + \alpha(2(k-j) - 1)$ and the second inequality is obtained using $\alpha \leq \frac{\phi}{j}$.

2. If $\frac{1}{j^2} \leq nd^2 < 1 \Rightarrow \frac{1}{j} \leq \sqrt{nd^2} \leq \alpha \leq \sqrt{nd^2}\phi$ because $\alpha = nd^2\frac{\sqrt{1 + \frac{4}{nd^2}} + 1}{2} = \sqrt{nd^2}\frac{\sqrt{nd^2+4} + \sqrt{nd^2}}{2}$, then we have that $n_j^{\rightleftharpoons k}$ satisfies

$$n_j^{\rightleftharpoons k} \geq n_j^{\rightarrow}\frac{n}{\alpha}\frac{(1+\alpha)^{2(k-j)} - 1}{(1+\alpha)^{2(k-j)} + 1} \geq n_j^{\rightarrow}\frac{n}{\alpha}\frac{(1 + \sqrt{nd^2})^{2(k-j)} - 1}{(1 + \sqrt{nd^2})^{2(k-j)} + 1}$$

where the second inequality follows because the ESS is monotonically increasing for $\alpha$ and $\alpha \geq nd^2$. Since $(1 + \sqrt{nd^2})^{2(k-j)} \geq 1 + 2\sqrt{nd^2}(k-j)$ and $k - j \geq 1$, we have that

$$n_j^{\rightleftharpoons k} \geq n_j^{\rightarrow} + \frac{n}{\alpha}\frac{\sqrt{nd^2}}{1 + \sqrt{nd^2}} \geq n_j^{\rightarrow} + n\frac{1}{\phi}\frac{1}{1 + \sqrt{nd^2}}$$

because $\alpha \leq \sqrt{nd^2}\phi$.

3. If $nd^2 \geq 1 \Rightarrow 1 \leq nd^2 \leq \alpha \leq nd^2\phi$ because $\alpha = nd^2\frac{\sqrt{1 + \frac{4}{nd^2}} + 1}{2}$, then we have that $n_j^{\rightleftharpoons k}$ satisfies

$$n_j^{\rightleftharpoons k} \geq n_j^{\rightarrow} + n\frac{1}{\alpha}\frac{2^{2(k-j)} - 1}{2^{2(k-j)} + 1} \geq n_j^{\rightarrow} + n\frac{1}{nd^2}\frac{1}{\phi}\frac{3}{5}$$

where the first inequality follows because the ESS is monotonically increasing for $\alpha$ and $\alpha \geq 1$ and the second inequality is obtained using $k - j \geq 1$ and $\alpha \leq nd^2\phi$.

$\square$

Table 2: Datasets characteristics.

| Dataset | Classes | Samples | Tasks |
|---|---|---|---|
| Yearbook | 2 | 37,921 | 126 |
| ImageNet Noise | 2 | 12,000 | 10 |
| DomainNet | 4 | 6,256 | 6 |
| UTKFace | 2 | 23,500 | 94 |
| Rotated MNIST | 2 | 70,000 | 60 |
| CLEAR | 3 | 10,490 | 10 |

## G    ADDITIONAL NUMERICAL RESULTS AND IMPLEMENTATION DETAILS

In this section we describe the datasets used for the numerical results in Section 5, we provide further details for the numerical experimentations carried out, and include several additional results. Specifically, in the first set of additional results, we evaluate the classification performance of the proposed method in comparison with state-of-the-art techniques for different sample sizes; in the second set of additional results, we further show the performance improvement leveraging information from preceding and succeeding tasks with additional datasets; in the third set of additional results, we show the classification error and the running time of LMRCs for different hyper-parameter values; and in the fourth set of additional results, we evaluate the assumption of change between tasks being independent and zero-mean. In addition, in the folder Implementation_LMRC in the supplementary materials we provide the code of the proposed LMRCs with the setting used in the numerical results.

The datasets used in Section 5 are publicly available in Ginosar et al. (2015); Russakovsky et al. (2015); Zhang et al. (2017); Peng et al. (2019); Lin et al. (2021), and http://yann.lecun.com/exdb/mnist/. The summary of these datasets is provided in Table 2 that shows the number of classes, the number of samples, and the number of tasks. In the following, we further describe the tasks and the time-dependency of each dataset used.

- The "Yearbook" dataset contains portraits' photographs over time and the goal is to predict males and females. Each task corresponds to portraits from one year from 1905 to 2013.

- The "ImageNet noise" dataset contains images with increasing noise over tasks and the goal is to predict if an image is a bird or a snake. The sequence of tasks corresponds to the noise factors $[0.0, 0.4, 0.8, 1.2, 1.6, 2.0, 2.4, 2.8, 3.2, 3.6]$ (Mai et al., 2022).

- The "DomainNet" dataset contains six different domains with decreasing realism and the goal is to predict if an image is an airplane, bus, ambulance, or police car. The sequence of tasks corresponds to the six domains: real, painting, infograph, clipart, sketch, and quickdraw.

- The "UTKFaces" dataset contains face images in the wild with increasing age and the goal is to predict males and females. The sequence of tasks corresponds to face images with different ages from 0 to 116 years.

- The "Rotated MNIST" dataset contains rotated images with increasing angles over tasks and the goal is to predict if the number in an image is greater than 5 or not. Each $j$-th task corresponds to a rotation angle randomly selected from $\left[\frac{180(j-1)}{k}, \frac{180j}{k}\right]$ degrees where $j \in \{1, 2, \ldots, k\}$ and $k$ is the number of tasks.

- The "CLEAR" dataset contains images with a natural temporal evolution of visual concepts in the real world and the goal is to predict if an image is soccer, hockey, or racing. Each task corresponds to one year from 2004 to 2014.

The samples in each task are randomly splitted in 100 samples for test and the rest of the samples for training. The samples used for training in the numerical results are randomly sampled from each group of training samples in each repetition.

The classifier parameters in the numerical results are obtained using an accelerated subgradient method based on Nesterov approach (Nesterov & Shikhman, 2015; Tao et al., 2019). Such subgradient method applied to optimization (3) obtains at each step classifier parameters $\boldsymbol{\mu}$ from the mean

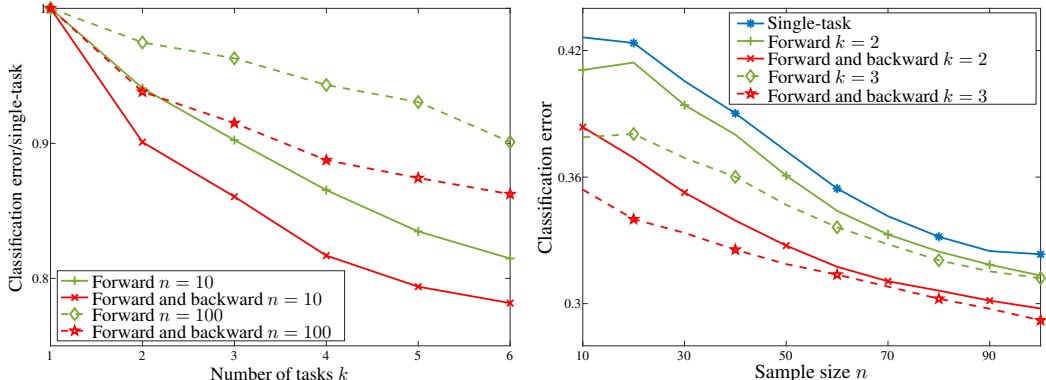

(a) Classification error per number of tasks using "DomainNet" dataset.

(b) Classification error per sample size using "DomainNet" dataset.

Figure 5: Forward and backward learning can sharply boost performance and ESS as tasks arrive.

Table 3: Classification error and standard deviation of the proposed LMRC method in comparison with the state-of-the-art techniques.

| Dataset | Yearbook | | ImageNet noise | | DomainNet | | UTKFaces | | Rotated MNIST | | CLEAR | |
|---|---|---|---|---|---|---|---|---|---|---|---|---|
| $n$ | 50 | 150 | 50 | 150 | 50 | 150 | 50 | 150 | 50 | 150 | 50 | 150 |
| GEM | .16 ± .02 | .16 ± .03 | .16 ± .06 | .12 ± .03 | .65 ± .05 | .49 ± .10 | .12 ± .00 | .12 ± .00 | .29 ± .04 | .27 ± .01 | .08 ± .01 | .08 ± .01 |
| MER | .11 ± .01 | .10 ± .01 | **.10** ± .01 | **.07** ± .01 | .29 ± .02 | .28 ± .02 | .11 ± .09 | .11 ± .01 | .41 ± .13 | .47 ± .05 | .09 ± .04 | .05 ± .02 |
| ELLA | .43 ± .10 | .43 ± .08 | .47 ± .05 | .47 ± .04 | .67 ± .05 | .67 ± .05 | .18 ± .11 | .17 ± .11 | .48 ± .05 | .47 ± .05 | .60 ± .05 | .60 ± .04 |
| EWC | .38 ± .02 | .22 ± .02 | .47 ± .05 | .45 ± .07 | .75 ± .05 | .74 ± .05 | .12 ± .00 | .12 ± .00 | .44 ± .01 | .38 ± .01 | .64 ± .03 | .60 ± .04 |
| LMRC | **.10** ± .02 | **.08** ± .01 | **.10** ± .02 | .08 ± .01 | **.28** ± .03 | **.27** ± .02 | **.10** ± .00 | **.10** ± .00 | **.25** ± .02 | **.20** ± .01 | **.05** ± .02 | **.04** ± .01 |

and confidence vectors $\boldsymbol{\tau}, \boldsymbol{\lambda}$ using the iterations for $l = 1, 2, \ldots, K$

$$\bar{\boldsymbol{\mu}}(l+1) = \boldsymbol{\mu}(l) + a_l\Big(\boldsymbol{\tau} - \partial\varphi(\boldsymbol{\mu}(l)) - \boldsymbol{\lambda}\mathrm{sign}(\boldsymbol{\mu}(l))\Big) \tag{23}$$

$$\boldsymbol{\mu}(l+1) = \bar{\boldsymbol{\mu}}(l+1) + \theta_{l+1}(\theta_l^{-1} - 1)\left(\boldsymbol{\mu}(l) - \bar{\boldsymbol{\mu}}(l)\right)$$

where $\mathrm{sign}(\cdot)$ denotes the sign function, $\boldsymbol{\mu}(l)$ is the $l$-th iterate for $\boldsymbol{\mu}$, $\theta_l = 2/(l+1)$ and $a_l = 1/(l+1)^{3/2}$ are the step sizes and $\partial\varphi(\boldsymbol{\mu}(l))$ denotes a subgradient of $\varphi(\cdot)$ at $\boldsymbol{\mu}(l)$ with

$$\varphi(\boldsymbol{\mu}) = \max_{x \in \mathcal{X}, \mathcal{C} \subseteq \mathcal{Y}} \frac{\sum_{y \in \mathcal{C}} \Phi(x,y)^{\mathrm{T}}\boldsymbol{\mu} - 1}{|\mathcal{C}|}.$$

In addition, the above subgradient method is implemented using $K = 2000$ iterations and a warm-start that initializes the classifier parameters in (23) with the solution obtained for the closest task.

In the first set of additional results, we further compare the classification error of LMRCs with the state-of-the-art techniques. The results in Table 1 in the paper as well as Table 3 are obtained computing the classification error 50 times for each sample size. Table 1 in the paper shows classification errors for $n = 10$ and $n = 100$ samples, while Table 3 shows the classification error for $n = 50$ and $n = 150$ samples. As can be observed in Table 3, the performance improvement of LMRCs in comparison with the state-of-the-art techniques for $n = 50$ and $n = 150$ is similar to that shown in the paper for $n = 10$ and $n = 100$.

In the second set of additional results, we further illustrate the relationship among classification error, number of tasks, and sample size. Figure 4 in the paper as well as Figure 5 are obtained computing the classification error over all the sequences of consecutive tasks of length $k$ in the dataset. Then, we repeat such experiment 10 times with randomly chosen training sets of size $n$. Figure 5 extends the results for LMRCs using "DomainNet" dataset completing those in the main paper that show the results using "Yearbook" dataset. Figure 5a shows the classification error of LMRC method divided by the classification error of single-task learning for different number of tasks with $n = 10$ and $n = 100$ sample sizes. In addition, Figure 5b shows the classification error of LMRC method for different sample sizes with $k = 10$ tasks. Figures 5a and 5b show similar behavior to Figures 4a and 4b in the paper, respectively. In addition, Figure 6 shows the classification error of LMRCs

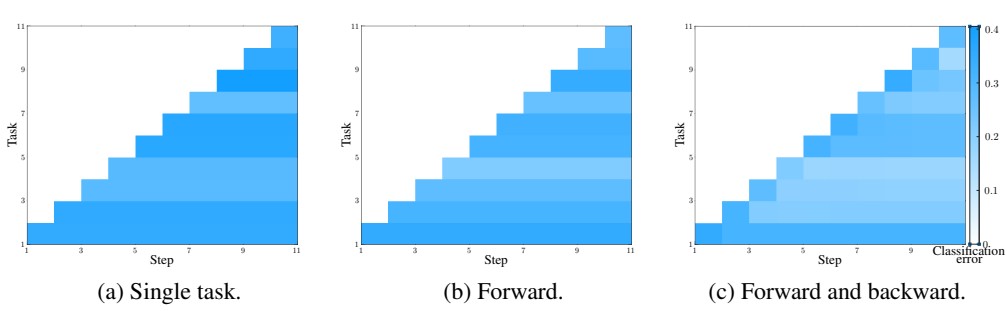

|  | (a) Single task. | (b) Forward. | (c) Forward and backward. |

Figure 6: Forward and backward learning can improve performance of preceding tasks.

Table 4: Classification error of the proposed LMRC method varying $W$ and $b$.

| Dataset | Yearbook | | ImageNet noise | | DomainNet | | UTKFaces | | Rotated MNIST | | CLEAR | |
|---|---|---|---|---|---|---|---|---|---|---|---|---|
| $n$ | 10 | 100 | 10 | 100 | 10 | 100 | 10 | 100 | 10 | 100 | 10 | 100 |
| $W = 2$ | .13 | .08 | .15 | .09 | .34 | .28 | .10 | .10 | .36 | .21 | .09 | .05 |
| $W = 4$ | .13 | .09 | .15 | .09 | .32 | .27 | .10 | .10 | .35 | .21 | .09 | .05 |
| $W = 6$ | .13 | .09 | .15 | .09 | .33 | .28 | .10 | .10 | .36 | .21 | .09 | .06 |
| $b = 1$ | .14 | .10 | .15 | .09 | .36 | .29 | .10 | .10 | .36 | .22 | .10 | .05 |
| $b = 2$ | .14 | .09 | .15 | .09 | .35 | .28 | .10 | .10 | .36 | .22 | .09 | .05 |
| $b = 3$ | .13 | .08 | .15 | .09 | .34 | .28 | .10 | .10 | .36 | .21 | .09 | .05 |
| $b = 4$ | .13 | .08 | .15 | .08 | .34 | .28 | .10 | .10 | .36 | .21 | .09 | .05 |
| $b = 5$ | .13 | .08 | .15 | .08 | .34 | .28 | .10 | .10 | .36 | .21 | .08 | .05 |

per step and task with single-task learning, forward learning, and forward and backward learning using the "Yearbook" dataset. Such figure shows that forward and backward learning can improve performance of preceding tasks, while forward learning and single task learning maintain the same performance over time.

In the third set of additional results, we further assess the change in classification error and the running time of LMRCs varying the hyper-parameters. Table 4 shows the classification error of LMRCs varying the values of hyper-parameter for the window size $W$ and the number of backward steps $b$, completing those in the paper that show the results for $W = 2$ and $b = 3$. As shown in the table, the proposed LMRCs do not require a careful fine-tuning of hyper-parameters and similar performances are obtained by using different values. In addition, Table 5 shows the mean running time per task in seconds of LMRCs for $b = 1, 2, \ldots, 5$ backward steps in comparison with the state-of-the-art techniques. Such table shows that the methods proposed for backward learning do not require a significant increase in complexity. In addition, Table 5 shows that the running time of the proposed method is similar to that of other state-of-the-art methods.

In the fourth set of additional results, we evaluate the assumption of change between tasks being independent and zero-mean by assessing the partial autocorrelation of mean vectors. In particular, the partial autocorrelation at any lag would be zero if tasks are i.i.d.; while the partial autocorrelation at lag 1 is larger than zero if tasks satisfy the assumption of Section 2. Figure 7 shows the averaged partial autocorrelation of the mean vectors components +/- their standard deviations for different lags using "Portraits" and "UTKFaces" datasets. Such figure shows a partial autocorrelation clearly non-zero at lag 1 that reflects dependence between consecutive mean vectors, as described by the assumption of Section 2."

Table 5: Running time in seconds of LMRC method in comparison with the state-of-the-art techniques.

| Dataset | Yearbook | | ImageNet noise | | DomainNet | | UTKFaces | | Rotated MNIST | | CLEAR | |
|---|---|---|---|---|---|---|---|---|---|---|---|---|
| $n$ | 10 | 100 | 10 | 100 | 10 | 100 | 10 | 100 | 10 | 100 | 10 | 100 |
| ELLA | 0.066 | 0.070 | 0.073 | 0.073 | 0.054 | 0.065 | 0.059 | 0.062 | 0.176 | 0.198 | 0.077 | 0.073 |
| GEM | 0.098 | 0.476 | 0.010 | 0.037 | 0.005 | 0.020 | 0.056 | 0.310 | 0.180 | 1.094 | 0.009 | 0.034 |
| MER | 0.166 | 3.726 | 0.079 | 1.052 | 0.066 | 0.900 | 0.127 | 3.458 | 0.211 | 4.092 | 0.074 | 1.063 |
| EWC | 0.358 | 3.031 | 0.032 | 0.252 | 0.018 | 0.155 | 0.245 | 2.246 | 0.587 | 5.296 | 0.031 | 0.248 |
| LMRC $b = 1$ | 0.094 | 0.324 | 0.259 | 0.490 | 0.518 | 8.463 | 0.108 | 0.348 | 0.135 | 0.471 | 0.235 | 1.310 |
| LMRC $b = 2$ | 0.105 | 0.397 | 0.261 | 0.531 | 0.543 | 8.983 | 0.115 | 0.401 | 0.165 | 0.599 | 0.252 | 1.406 |
| LMRC $b = 3$ | 0.133 | 0.487 | 0.284 | 0.561 | 0.542 | 9.514 | 0.133 | 0.488 | 0.209 | 0.737 | 0.255 | 1.469 |
| LMRC $b = 4$ | 0.167 | 0.582 | 0.304 | 0.585 | 0.559 | 9.699 | 0.156 | 0.573 | 0.249 | 0.877 | 0.271 | 1.595 |
| LMRC $b = 5$ | 0.184 | 0.664 | 0.438 | 0.601 | 0.571 | 9.921 | 0.190 | 0.664 | 0.288 | 1.010 | 0.360 | 1.693 |

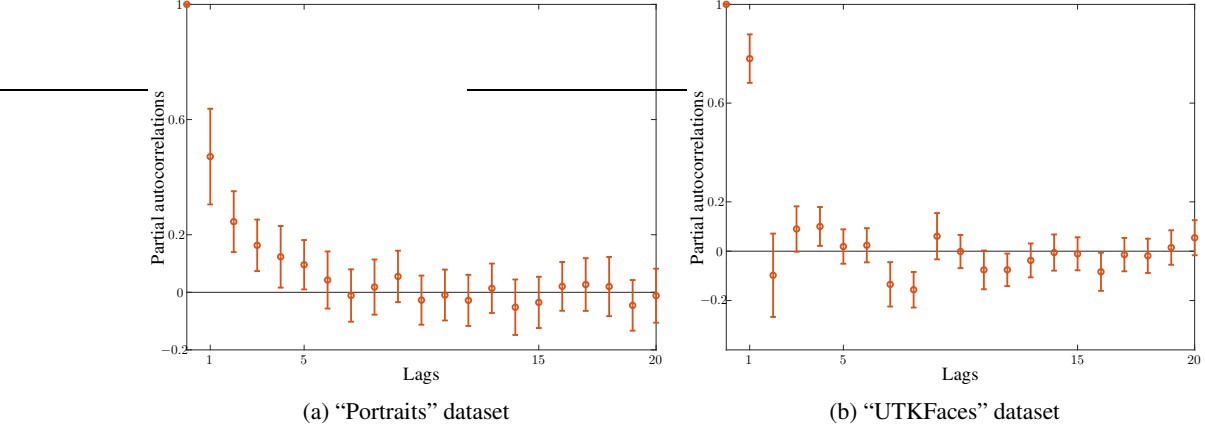

(a) "Portraits" dataset  (b) "UTKFaces" dataset

Figure 7: Averaged partial autocorrelation of mean vectors components +/- their standard deviations.

