# OpenReview forum: "Forward and Backward Lifelong Learning with Time-dependent Tasks"
_ICLR.cc/2023/Conference — Submitted to ICLR 2023_

### Official Review · Reviewer_vTda · 2022-10-18

**Confidence:** 2
**Correctness:** 2
**Technical Novelty And Significance:** 3
**Empirical Novelty And Significance:** 2
**Recommendation:** 5

**Clarity, Quality, Novelty And Reproducibility:**

The paper is well-written in general. However, the underlying logic may not be clear to some audiences; it's not easy to figure out the big picture. Because of that clarity issue and the lacking of comparisons with existing methods, it's hard to evaluate the novelty. The reproducibility is satisfactory.

**Strength And Weaknesses:**

Strength.

(1) The paper is well-written.

(2) The presented theorems and derivations seem to be solid.

Weaknesses.

(1) The related work and the baseline methods in the experiments are kind of outdated.

(2) It seems the presented method assumes one can access all $\tau_i, i \in [1,j+d]$ when dealing with the $j$-th task.

(3) It seems the presented method is presented based on a frozen pre-trained feature extractor and simple Gaussian assumptions (\ie Eqs (5-6) and (10-11) from the Kalman filtering).

**Summary Of The Paper:**

This paper presents a method based on minimax risk classifiers, termed LMRCs, for lifelong learning with time-dependent classification tasks. The LMRCs exploits forward and backward knowledge transfer based on underlying Gaussian assumptions, with perhaps a frozen pre-trained feature extractor. Theoretical analyses and empirical experimental results are presented.

**Summary Of The Review:**

There are so many lifelong learning papers in recent years; please make comparisons and discuss the advantages of the proposed method.

How to train the feature mapping $\Phi(x,y)$ with the presented LMRCs? Is $\Phi(x,y)$ assumed known/frozen?

In the paragraph before Eq (3), what's the 0-1-loss? Does that mean you assume 2-category classification tasks throughout the paper?

In Eqs (4-6), the derivations are based on Gaussian assumptions, right? How do those assumptions affect the forward/backward knowledge transfer in lifelong learning?

In the experiments, the baseline methods are kind of outdated.

---

> ### Author Response · Authors · 2022-11-16
> **Response to Reviewer vTda**
>
> Q1. It seems the presented method assumes one can access all $\boldsymbol{\tau}_i, i\in [1, j+d]$ when dealing with the $j$-th task.
>
> R1. In the new version of the paper, we have clarified in Section 4 that at step $k$ the methods proposed access mean vectors $\boldsymbol{\tau}_i \in \mathbb{R}^m, i \in [k-b+1, k]$ for the last $b$ tasks of the sequence of $k$ tasks (see also updated Algorithm 1). Specifically, with forward and backward learning, the methods proposed learn a classification rule for each $j$-th task at step $j$ and then update such rule during the next $b-1$ steps, so that at each step $k$ the proposed method access the last $b$ mean vectors, $\boldsymbol{\tau}_i, i \in [k-b+1, k]$. The theoretical results in Section 4 shows that the benefits of backward learning are achieved using a reduced value for $b$ such as $b = 3$. Numerical results in Section 5 shows the classification error using $b = 3$ and Appendix G shows the classification error varying the number of backward steps $b$.
>
> Q2. It seems the presented method is presented based on a frozen pre-trained feature extractor and simple Gaussian assumptions (ie Eqs (5-6) and (10-11) from the Kalman filtering). In Eqs (4-6), the derivations are based on Gaussian assumptions, right? How do those assumptions affect the forward/backward knowledge transfer in lifelong learning?
>
>
>
> R2. In the updated Appendix A, we have clarified that recursions (5)-(6) and (10)-(11) in Sections 3 and 4 do not require any distributional assumption to provide the unbiased linear estimator with the minimum MSE together with its MSE. Recursions in (5)-(6) and (10)-(11) are given by the Kalman filtering and smoothing recursions corresponding to the linear dynamical system in (14). Such recursions provide the unbiased linear estimator of mean vector $\boldsymbol{\tau}^\infty$ with minimum MSE as long as $\boldsymbol{v}_j$ for $j \in \{1, 2, ..., k\}$ are independent and zero mean, and independent of $\boldsymbol{w}_j$ for $j \in \{1, 2, ..., k\}$, which is satisfied because $\mathrm{p}_\{j+1}-\mathrm{p}_j$ are zero mean and independent for $j \in \{1, 2, ..., k-1\}$. If $\boldsymbol{w}_j$ and $\boldsymbol{v}_j$ in (14) and (15) were Gaussian distributed, the Kalman filtering and smoothing recursions would provide an estimator with even stronger theoretical properties but such Gaussian assumptions are not required for the results shown in the paper.
>
> Q3. How to train the feature mapping $\Phi(x, y)$ with the presented LMRCs? Is $\Phi(x, y)$ assumed known/frozen?
>
> R3. LMRCs use a pre-defined feature mapping for all tasks in the sequence as is done in other methods for lifelong learning, e.g., Kemker \& Kanan, (2018); Hurtado et al., (2021). As described in Figures 3 and 4 and in Section 5, the techniques proposed can boost the ESS of each task specially in situations with few examples per task. In these situations learning an adequate feature mapping for each task would be very difficult.
>
> Q4. In the paragraph before Eq (3), what's the 0-1-loss? Does that mean you assume 2-category classification tasks throughout the paper?
>
> R4. In the paper we use the 0-1-loss, which quantifies the classification error for a general number of clases as
>
> $$\ell(\mathrm{h}, (x, y)) = \left\\{\begin{matrix}0 & \text{ if } & y = \mathrm{h}(x)\\\\
> 1 & \text{ if } & y \neq \mathrm{h}(x) \end{matrix}\right.$$
> where $\mathrm{h}(x) \in \mathcal{Y}$ denotes the label that assigns the rule $\mathrm{h}$ to the instance $x \in \mathcal{X}$. In the updated paper, we have clarified in Section 2 that the expected loss using 0-1-loss is the probability of classification error so that expected losses and performance guarantees in Theorems 1 and 3 are easier to interpret.

---

### Official Review · Reviewer_KxHd · 2022-10-26

**Confidence:** 4
**Correctness:** 3
**Technical Novelty And Significance:** 3
**Empirical Novelty And Significance:** 3
**Recommendation:** 6

**Clarity, Quality, Novelty And Reproducibility:**

I have the following comments/questions. I look forward to the response/clarification from the author(s). Thanks.

1. For a better understanding of the theoretical analyses, such as Theorems 1 and 2, some empirical experiments may be needed for further verification. In addition, are these bounds tight?

2. The theoretical analyses in this paper only hold when the loss function is 0-1 loss, right? Or can it be applied to more general or arbitrary loss functions?


In addition, some tiny issues/typos:

(1) The first line below Eq. (1), "$\ell$(h,p) denotes the expected loss of classification...". To be consistent with the above, here it should be "error probability", right?

(2) Some abbreviations and full names appear many times throughout the main text. This is not necessary.

(3) The format of the references is inconsistent. Please check carefully and correct it.

---




**Strength And Weaknesses:**

### Strength:

1. Trying to address the issues of catastrophic forgetting and i.i.d. constraint in current lifelong learning.

2. Providing theoretical guarantees for the proposed algorithm.

3. Performing empirical experiments to validate the performance of the proposed algorithm.

---

### Weaknesses:

1. The theoretical analysis of the proposed algorithm, such as generalization bounds, may require further empirical experiments.

2. Some descriptions in this paper are not so clear.

For more details, please see the section of "Clarity, Quality, Novelty And Reproducibility".

---

**Summary Of The Paper:**

In this paper, the author(s) used minimax risk classifiers to propose a lifelong method that utilized forward and backward learning for time-dependent tasks. Through Empirical experiments, the author(s) tried to show the advantage of the proposed algorithm.

---

**Summary Of The Review:**

I think the topic in this paper is important and this paper is generally well-written; However, there are some unclear aspects in the paper. Maybe it needs the author(s) to clarify them. Thanks.

---

> ### Author Response · Authors · 2022-11-16
> **Response to Reviewer KxHd**
>
> Q1. For a better understanding of the theoretical analyses, such as Theorems 1 and 2, some empirical experiments may be needed for further verification. In addition, are these bounds tight?
>
> R1. Theorems 1 and 3 bound the minimax risk for each task. As shown in Mazuelas et al., (2022a), such minimax risk can provide tight performance guarantees for the expected loss of the corresponding minimax classifier. In particular, such results show that in standard supervised classification the bounds decrease for increasing sample sizes $n$ at a rate $\mathcal{O}(1/\sqrt{n})$. The goal of Theorems 1 and 3 in the paper is to show that, using the methods proposed, the corresponding bounds decrease at a rate given by the ESS, that is, $\mathcal{O}(1/\sqrt{ESS})$ instead of $\mathcal{O}(1/\sqrt{n})$. In addition, such theorems together with Theorems 2 and 4 quantify how much larger the ESS is compared with the sample size $n$. In particular, the theoretical results show that backward learning results in higher ESS, and that the ESS increases with the number of tasks. Such qualitative consequences of the theoretical results are corroborated by the numerical results in Section 5 and Appendix G, for instance in Figures 4 and 5.
>
> Q2. The theoretical analyses in this paper only hold when the loss function is 0-1 loss, right? Or can it be applied to more general or arbitrary loss functions?
>
> R2. The framework of MRCs can be utilized and theoretically analyzed with general loss functions, as shown in Mazuelas et al. (2022b). The methodology presented for lifelong learning could be used with general loss functions analogously as is done in Mazuelas et al. (2022b) for standard supervised classification.  In this paper, we utilize 0-1-loss so that its expected loss (error probability) is easier to interpret.
>
> Q3. The first line below Eq. (1), "$\ell$(h,p) denotes the expected loss of classification\ldots". To be consistent with the above, here it should be "error probability", right?
>
> R3. Thanks for pointing out this posible source of confusion. In the updated version of the paper, we have clarified at the beginning of Section 2 that $\ell(\mathrm{h}, \mathrm{p})$ denotes the expected loss of classification rule $\mathrm{h}$ for distribution $\mathrm{p}$, then we point out that in the paper we focus on 0-1-loss so that expected losses become error probabilities.
>
> Q4. Some abbreviations and full names appear many times throughout the main text. This is not necessary. The format of the references is inconsistent. Please check carefully and correct it.
>
> R4. We thank the Reviewer for finding out those typos. We have fixed them in the new version of the paper.

---

### Official Review · Reviewer_Q8gE · 2022-10-31

**Confidence:** 3
**Correctness:** 3
**Technical Novelty And Significance:** 3
**Empirical Novelty And Significance:** 2
**Recommendation:** 5

**Clarity, Quality, Novelty And Reproducibility:**

The quality and clarity of the paper are good but could be improved. For example,
1. I would suggest writing the assumption of $p_{i+1}-p_i$ being independent and zero-mean with formal statement. For example, is the zero-mean of it the measure that maps all sigma-algebra to zero?
2. At the beginning of section 3.2, it says "we denote by $R^\\infty_j$ the smallest minimax risk ". How is the "smallest minimax risk" defined, e.g., under which uncertainty set?

The proposed method and its analysis are novel in the sense that the authors adopt minimax risk classifiers (MRC) [Mazuelas et al.],

**Strength And Weaknesses:**

Strength: This paper presents an interesting framework for lifelong learning, i.e., the lifelong minimax risk classifiers (LMRCs), and the proposed method is novel to me. The theoretical analysis seems solid.

Weakness: I'm concerned about a few settings and assumptions where more justification is needed.
1. The assumption of $p_{i+1}-p_i$ being independent and zero-mean is only justified by a sentence. However, the validness of such assumption should be discussed more in my opinion, since it is an important assumption to the proposed method. For example, are there any real-life settings where this assumption may hold true (or even approximately being true).
2. The proposed method relies on the feature map $\\Phi$ which is not learnable. However, one would argue that without learning the feature map it is not really doing a proper lifelong learning.
3. Instead of learning the feature map, the proposed method learns the uncertainty set. How would one know if the leaned uncertainty set is a good one? For example, will the uncertainty set be too big such that the maximization part overpowers the minimization part, and the resulting classifier is not really meaningful? Alternatively, does the method provide a bound on the regular loss function for each task?

**Summary Of The Paper:**

This paper tackles lifelong learning by improve effective sample size using both forward and backward tasks. This paper assumes that tasks arrive consecutively non-iid but satisfying a martingale assumption. This paper adopts the framework minimax risk classification into the lifelong learning setting. After showing the improvement in effective sample size theoretically, the paper presents numerical experiments to demonstrate the effectiveness of the proposed method.

**Summary Of The Review:**

Although the paper is interesting and somewhat novel, I am concerned about a few settings used/assumed in the paper, as detailed in the weakness section.

---

> ### Author Response · Authors · 2022-11-16
> **Response to Reviewer Q8gE (1/2)**
>
> Q1. The assumption of $\mathrm{p}_{i+1}-\mathrm{p}_i$ being independent and zero-mean is only justified by a sentence. However, the validness of such assumption should be discussed more in my opinion, since it is an important assumption to the proposed method. For example, are there any real-life settings where this assumption may hold true (or even approximately being true).
>
> R1. In the new version of the paper, we have further described the assumption of $\mathrm{p}_{i+1}-\mathrm{p}_i$ being independent and zero-mean. Specifically, the updated Section 2 provides more detailed justification of the assumption and new Appendix G supports that the assumption more accurately describe datasets with time-dependent tasks than the usual i.i.d. assumption.
>
> In Section 2, we have included the following comments:
>
> ``In the following, we propose lifelong learning techniques designed for time-dependent tasks that are characterized by distributions $\mathrm{p}_1, \mathrm{p}_2, ...$ such that the changes between consecutive distributions $\mathrm{p}_\{i+1} - \mathrm{p}_i$ are independent and zero-mean random signed measures for $i = 1, 2, ...$. Such assumption can account for usual higher similarities between consecutive tasks; for instance, it implies that $\mathrm{p}_\{i+t} - \mathrm{p}_i$ is a zero-mean random variable with $\mathbb{V}\text{ar}\\{\mathrm{p}_\{i+t}-\mathrm{p}_i\\}=\sum_\{j=1}^t \mathbb{V}\text{ar}\\{\mathrm{p}_\{i+j}-\mathrm{p}_\{i+j-1}\\}$,
> while the i.i.d. case would imply that $\mathrm{p}_\{i+t} - \mathrm{p}_i$ is a zero-mean random variable with $\mathbb{V}\text{ar}\\{\mathrm{p}_\{i+t}-\mathrm{p}_i\\}=\mathbb{V}\text{ar}\\{\mathrm{p}_\{i+1}-\mathrm{p}_i\\} = 2\mathbb{V}\text{ar}\\{\mathrm{p}_1\\}$ for any $t$ and $i$.''
>
>
> The assumption in the paper describes scenarios in which the underlying distributions change as a random walk with independent increments, so that it can more accurately describe time-dependent tasks. For instance, such type of assumption is often used to describe time-dependent processes such as stock prices. The assumption in the paper differs from the usual i.i.d. assumption but is not necessarily stronger. Note that both with the random walk assumption and with the i.i.d. assumption, $\mathrm{p}_\{i+1}-\mathrm{p}_i$ are random signed measures with mean the zero measure. The main difference between both assumptions is that with the random walk assumption each difference between consecutive distributions $\mathrm{p}_\{i+1}-\mathrm{p}_i$ is independent of the others, while with the i.i.d. assumption each distribution $\mathrm{p}_i$ is independent of the others.
>
> In Appendix G, we have included numerical results showing that the assumption in the paper can better describe datasets with time-dependent tasks than the usual i.i.d. assumption. In particular, we have included Figure 7 and the following comments.
>
> ''In the fourth set of additional results, we evaluate the assumption of change between tasks being independent and zero-mean by assessing the partial autocorrelation of mean vectors. In particular, the partial autocorrelation at any lag would be zero if tasks are i.i.d.; while the partial autocorrelation at lag 1 is larger than zero if tasks satisfy the assumption of Section 2. Figure 7 shows the averaged partial autocorrelation of the mean vectors components +/- their standard deviations for different lags using ``Portraits” and “UTKFaces” datasets. Such figure shows a partial autocorrelation clearly non-zero at lag $1$ that reflects dependence between consecutive mean vectors, as described by the assumption of Section 2.''
>
> Partial autocorrelations are the usual tool to assess if a process is a random walk (see Section 4 in Cowpertwait and Metcalfe, (2009)). In particular, the partial autocorrelation at any lag would be zero if tasks are i.i.d.; while the partial autocorrelation at lag 1 is larger than zero if the sequence of tasks is a random walk. Partial autocorrelations in Figure 7 of Appendix G show dependences of mean vectors for different lags. Note that if the distributions satisfy the i.i.d. (resp. random walk) assumption, then mean vectors satisfy the i.i.d (resp. random walk) assumption. Figure 7 shows a partial autocorrelation clearly non-zero at lag $1$ that strongly supports the random walk assumption instead of the i.i.d. assumption.
>
> Paul SP Cowpertwait and Andrew V. Metcalfe. *Introductory time series with R.* Springer Science \& Business Media, 2009.

---

> ### Author Response · Authors · 2022-11-16
> **Response to Reviewer Q8gE (2/2)**
>
> Q2. The proposed method relies on the feature map $\Phi$ which is not learnable. However, one would argue that without learning the feature map it is not really doing a proper lifelong learning.
>
> R2. LMRCs use a pre-defined feature mapping for all tasks in the sequence as is done in other methods for lifelong learning, e.g., Kemker \& Kanan, (2018); Hurtado et al., (2021). As described in Figures 3 and 4 and in Section 5, the techniques proposed can boost the ESS of each task specially in situations with few examples per task. In these situations learning an adequate feature mapping for each task would be very difficult.
>
> Q3. Instead of learning the feature map, the proposed method learns the uncertainty set. How would one know if the leaned uncertainty set is a good one? For example, will the uncertainty set be too big such that the maximization part overpowers the minimization part, and the resulting classifier is not really meaningful? Alternatively, does the method provide a bound on the regular loss function for each task?
>
> R3. The uncertainty set $\mathcal{U}$ is good if the minimax risk $R(\mathcal{U})$ obtained at learning is small because such minimax risk determines the error of MRCs in the worst-case. The smallest minimax risk corresponds to the ideal case of knowing the mean vectors exactly and Theorems 1 and 3 show the difference between the minimax risk of the uncertainty set used and the smallest minimax risk. In standard supervised classification, such difference is of the order $\mathcal{O}(1/\sqrt{n})$ with $n$ the sample size, while using the methods proposed, such difference  is of the order $\mathcal{O}(1/\sqrt{ESS})$, as shown in Theorems 1 and 3. Such theorems together with Theorems 2 and 4 quantify how much larger the ESS is compared with the sample size $n$.
>
> The framework of MRCs can be utilized and theoretically analyzed with general loss functions, as shown in Mazuelas et al. (2022b). The methodology presented for lifelong learning could be used with general loss functions analogously as is done in Mazuelas et al. (2022b) for standard supervised classification. In this paper, we utilize 0-1-loss so that its expected loss (error probability) is easier to interpret.
>
> Q4. I would suggest writing the assumption of $\mathrm{p}_{i+1} - \mathrm{p}_i$ being independent and zero-mean with formal statement. For example, is the zero-mean of it the measure that maps all sigma-algebra to zero?
>
> R4. In Section 2, we have more formally described the assumption as follows.
>
> ''Most existing lifelong learning techniques are designed for tasks characterized by distributions $\mathrm{p}_1, \mathrm{p}_2, ...$ such that the tasks’ distributions $\mathrm{p}_i$ are independent and identically distributed (i.i.d.) random probability measures for $i = 1, 2, ...$. In the following, we propose lifelong learning techniques designed for time-dependent tasks that are characterized by distributions $\mathrm{p}_1, \mathrm{p}_2, ...$ such that the changes between consecutive distributions $\mathrm{p}_\{i+1} - \mathrm{p}_i$ are independent and zero-mean random signed measures
> for $i = 1, 2, ...$.''
>
> Q5. At the beginning of section 3.2, it says "we denote by $R_j^\infty$ the smallest minimax risk ". How is the "smallest minimax risk" defined, e.g., under which uncertainty set?
>
> R5. In the new version of the paper, we have further described the smallest minimax risk $R^\infty$ in Sections~2 and 3.2. The smallest minimax risk $R^\infty$ corresponds with the uncertainty set $\mathcal{U}^\infty = \\{\mathrm{p} \in \Delta(\mathcal{X}\times\mathcal{Y}):\mathbb{E}_\{\mathrm{p}}\\{\Phi(x, y)\\} = \mathbb{E}_\{\mathrm{p}^*}\\{\Phi(x, y)\\} = \boldsymbol{\tau}^\infty\\}$ given by the true expectation of the feature mapping $\Phi$.

---

### Decision · Program_Chairs · 2023-01-20

**Decision:**

Reject

**Justification For Why Not Higher Score:**

The paper does not have a strong set of baselines to compare the proposed algorithm so it is not clear if it is doing better than the existing methods.

**Justification For Why Not Lower Score:**

N/A

**Metareview: Summary, Strengths And Weaknesses:**

This paper proposes a new algorithm for lifelong learning called LMRC which is based on the idea of minimax risk classifiers. The paper has a strong theoretical analysis of the proposed algorithm. However, the main weakness of this paper is the lack of strong baselines. Lifelong learning is an active research area with numerous algorithms published in the last few years. However, the latest baseline in the paper is from 2018. The paper also needs a strong literature review to position this algorithm when compared to other lifelong learning solutions. I recommend rejection and encourage the authors to implement stronger baselines, add a stronger related work section and resubmit to a future venue.